# Land use and land cover change in a tropical mountain landscape of northern Ecuador: Altitudinal patterns and driving forces

**Paulina Guarderas**[1,2]*, **Franz Smith**[3], **Marc Dufrene**[2]

1 Facultad de Ciencias Biológicas, Universidad Central del Ecuador, Quito, Ecuador, 2 Biodiversity and Landscape, TERRA, Teaching and Research Centre, Gembloux Agro-Bio Tech, University of Liège, Gembloux, Belgium, 3 Colegio de Ciencias Biológicas y Ambientales, Universidad San Francisco de Quito, Quito, Ecuador

* apguarderas@uce.edu.ec

**Data Availability Statement:** All relevant data are available in the following public repository: DOI 10.5281/zenodo.5911876.

## Abstract

Tropical mountain ecosystems are threatened by land use pressures, compromising their capacity to provide ecosystem services. Although local patterns and interactions among anthropogenic and biophysical factors shape these socio-ecological systems, the analysis of landscape changes and their driving forces is often qualitative and sector oriented. Using the Driver-Pressure-State-Impact-Response (DPSIR) framework, we characterized land use land cover (LULC) dynamics using Markov chain probabilities by elevation and geographic settings and then integrated them with a variety of publicly available geospatial and temporal data into a Generalized Additive Model (GAM) to evaluate factors driving such landscape dynamics in a sensitive region of the northern Ecuadorian Andes. In previous agricultural land located at lower elevations to the east of the studied territory, we found a significant expansion of floriculture (13 times) and urban areas (25 times), reaching together almost 10% of the territory from 1990 to 2014. Our findings also revealed an unexpected trend of páramo stability (0.75–0.90), but also a 40% reduction of montane forests, with the lowest probability (<0.50) of persistence in the elevation band of 2800–3300 m; agricultural land is replacing this LULC classes at higher elevation. These trends highlight the increasing threat of permanently losing the already vulnerable native mountain biodiversity. GAMs of socio-economic factors, demographic, infrastructure variables, and environmental parameters explained between 21 to 42% of the variation of LULC transitions observed in the study region, where topographic factors was the main drivers of change. The conceptual and methodological approach of our findings demonstrate how dynamic patterns through space and time and their explanatory drivers can assist local authorities and decision makers to improve sustainable resource land management in vulnerable landscapes such as the tropical Andes in northern Ecuador.

**Funding:** This research was funded by the Research Department and the Faculty of Biological Sciences at Universidad Central del Ecuador (Grant N. DOCT-DI-19-05); along with financial support from the Académie de Recherche et d'Enseignement Supérieur (ARES) from Belgium to P.G.

**Competing interests:** The authors have declared that no competing interests exist.

## Introduction

Tropical mountain systems supply vital benefits to millions of upland and lowland inhabitants [1] through the provision of Ecosystem Services (ES) [2, 3] and represent a global hotspot of tropical biodiversity and habitat refugia [4]. These areas are increasingly being transformed by human activities [4, 5]. Although the human activities in this region, including intensive traditional agriculture, have impacted its history of landscape patterns for centuries [5], recent transitions have also been documented [6–9], changes to this landscape's natural cycles and heterogeneity is reducing the capacity of the system to provide multiple benefits to people and guarantee their long-term sustainability [5].

Deforestation and agricultural intensification are the dominant transitions in many Andean systems [8]. However, forest recovery due to agricultural de-intensification and transitions between crops, pastures, and secondary vegetation, in addition to urban and agro-industrial expansion have also been observed in these systems. In-depth multi-temporal change studies are required to better understand this complexity in order to balance biodiversity conservation with human needs [6, 8, 10].

These distinct patterns of land utilization by various human activities (land use), in addition to spatial changes of biophysical cover on the earth's surface (land cover) [11] observed in the Tropical Andes vary with demographic, socio-economic, cultural and technological factors [8, 12, 13]. Additionally, these drivers interact with biophysical features like elevation, topography, soil and climate parameters, operating across spatial, temporal, and organizational scales [3, 14, 15]. For example, increasing global demand for food and non-food crops can drive agriculture expansion onto more fertile and flat land [6, 14, 16], whereas natural ecosystem recovery has been observed in abandoned marginal agricultural land [5, 17, 18].

Despite the documented useful insights into how different drivers can influence Land Use Land Cover (LULC) change in tropical mountain systems [6, 7, 10, 13], evidence from synthetical studies suggests that no universal link between cause and effect exists to explain deforestation and other LULC changes [5, 6, 14]. Different combinations of various proximate causes and underlying driving forces in varying geographical and historical contexts could affect landscape changes [8, 10].

Understanding future changes in tropical mountain systems and their associated ES relies on ecosystem assessments to document LULC pattern dynamics across environmental gradients and different temporal scales [19, 20]. Additionally, revealing interactive effects of distinct anthropogenic influences on landscape dynamics will be valuable for informing management [5], given the high vulnerability to climate change of highland landscapes like the Ecuadorian Andes [21]. Conducting integrated ecosystem assessments for adaptive management is urgently needed in highland tropical ecosystems where biodiversity conservation, sustainable use of natural resources, and the supply of essential ES should be assured [8, 22].

The Driver-Pressure-State-Impact-Response (DPSIR) framework links cause-effect relationships and feedback between human and natural systems [23] to understand and sustainably manage environmental problems [24]. Within the DPSIR framework, the anthropogenic impacts on ecosystems and their services can be described by social, demographic, economic, and other biophysical driving forces where these drivers exert pressures on the environment, affecting the state and condition of ecosystems [25]. Understanding this complexity is fundamental for the development of policies and measures for landscape planning and management, as societal responses to overcome environmental impacts [26].

Within this context, our study is unique in that it adapts the DPSIR holistic approach to the context of tropical mountain systems and implements the first elements of the framework to further complete an ES assessment in a sensitive region of the northeastern Ecuadorian Andes.

The study region comprises a landscape with distinct climatic conditions and management regimes along its elevation gradient, where floriculture crops and urban centers are emerging in an agricultural matrix, posing more pressure on remnant native ecosystems and their services.

Specifically, we addressed two questions: (1) what are the LULC change patterns across geographical and biophysical settings, in terms of the rate, magnitude, and direction of those changes, emphasizing trends in native ecosystems as sentinel habitats, and (2) what combination of environmental and anthropic factors can best explain the different landscape transitions.

## Materials and methods

### Conceptual framework

The DPSIR framework has been widely applied in ecosystem assessments to evaluate the impact of environmental changes on human well-being [3, 25, 27, 28]. Furthermore, since ecosystem assessments are based on scientific evidence, they are considered key management tools for decision making processes and adaptive management at landscape scales [29].

In the context of mountain systems, the DPSIR framework was initially conceptualized and implemented by Oddermat [30]. Recent initiatives have implemented this conceptual model for evaluating the state of mountain systems in distinct regions [29], but an adaptation of such an approach to conduct ecosystem assessments was lacking for the tropical mountain system context [31].

In this study, we adapted the DPSIR framework, by [24, 25, 27], to identify the key characteristics of tropical mountain systems that should be represented in ecosystem assessments at a landscape scale (Fig 1). In this context, driving forces will exert pressures, changing the state of

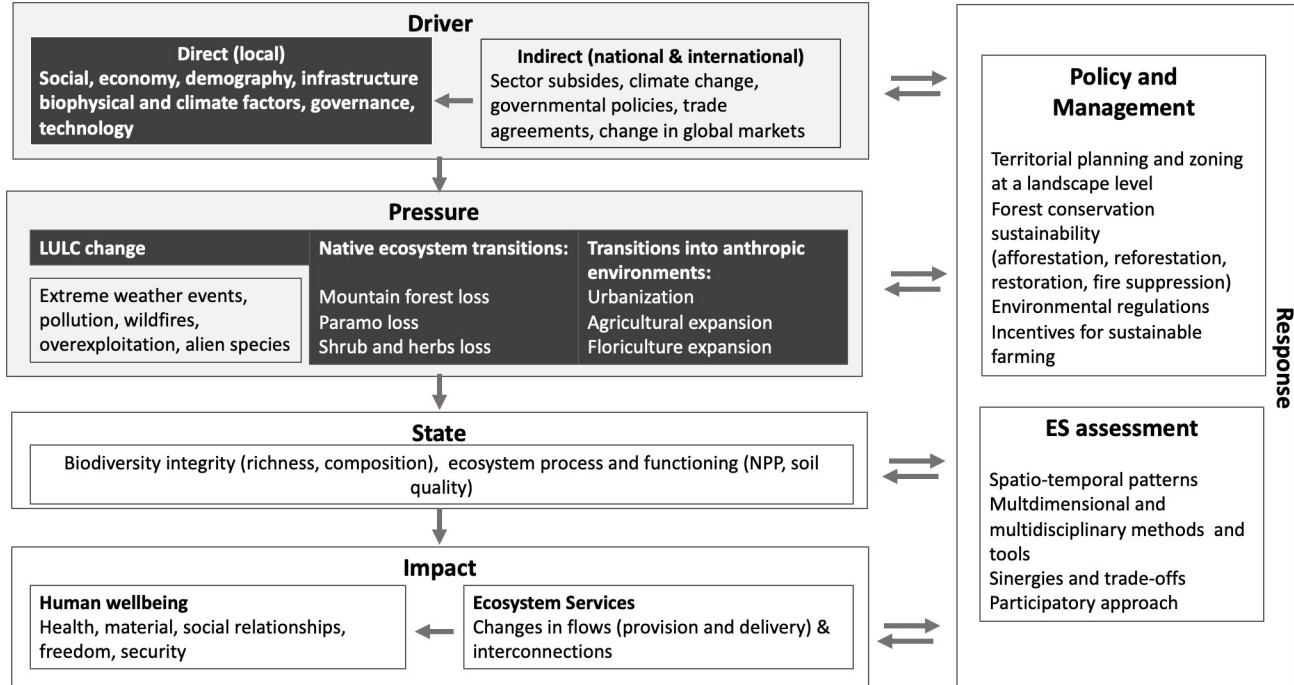

**Fig 1. DPSIR framework for ecosystem assessments in tropical mountain systems.** Arrows indicate causal relationships between driver, pressure, state, impact, and response (Adapted from [24, 25, 27]).

the system [29]. This altered state could ultimately impact on human wellbeing and lead to a societal response. The societal response in turn feeds back to all other components [24].

Drivers are the underlying causes of environmental change, and we consider that both direct and indirect driving forces are shaping mountain landscapes in the tropics (Fig 1). Indirect drivers act by modifying the conditions of one or more direct drivers, while direct drivers explicitly influence the system [24]. We integrated the scale of the impact into the level of influence of the driving forces as follows: the direct drivers are considered as local forces (such as demographic, economic, cultural, socio-political, governance and technological factors) (Fig 1), while the indirect drivers were forced as exogenous or external factors which operate at larger scales [29]. Then, indirect driving forces could include sector subsidies, government policies, trade agreements, change in global markets, and even climate change (Fig 1). Within the direct drivers, we considered it important to add the governance dimension, as proposed by [31], to complement the DPSIR framework with the holistic conceptualization of the Socio-Ecological Framework (SEF) and to analyze the interaction between social and ecosystem processes [32].

Pressure is the result of the interacting driving forces and generally represents a measurable human induced effect on the system–such as land use change, extreme weather events, pollution, wildfires, and overexploitation [24, 29]. In this article, we evaluated LULC change as the pressure element in the DPSIR approach. We operationalized LULC change considering two main landscape transitions: 1) the loss of native ecosystems and 2) the conversion to anthropic environments (Fig 1).

Pressures on the environment as a consequence of the driving forces could impact the state of the system. Here we described the state of tropical mountain systems in terms of their unique and vulnerable taxonomic and functional biodiversity (e.g., richness, composition, trophic groups) and their derived ecosystem properties (e.g., primary productivity, soil quality, vegetation cover, etc.) (Fig 1) [25].

Likewise, changes in the state of ecosystems impact on the provision and flow of ecosystem services and the associated benefits on people's quality of life (Fig 1). Tropical mountain systems are characterized by their contribution to essential ecosystem services such as water and food provision, carbon sequestration, landslide and erosion prevention, microclimate regulation, and the provision of multiple cultural services [33]. The level at which the provision of ecosystem services changes as a result of environmental changes will also impact on the wellbeing of people [25].

The final step in the DPSIR framework corresponds to the response component, which is envisioned as the societal acknowledgment of the state of the system and their feedback to overcome the impacts due to human activities [29]. According to [27], in the DPSIR framework the responses could be disaggregated into: 1) ES assessments and 2) policy and management where these aspects have been integrated into the DPSIR approach for this study (Fig 1). ES assessments should encompass multiple dimensions and disciplines to understand synergies, trade-offs, and interconnections of ES. These ES assessments should include participatory approaches and their scope should characterize geographical, biophysical, and temporal patterns. In tropical mountain systems, local and medium levels of territorial governance are key elements to implement policy and manage responses to overcome environmental issues. For instance, territorial planning and zoning schemes could organize a more balanced and multi-functional system of ES provision and flow at landscape scales. In addition, initiatives for native forest sustainability could be fostered at the local and medium levels of governance. Environmental regulations and incentives for the sustainable use of natural resources could also be supported at the national level of governance.

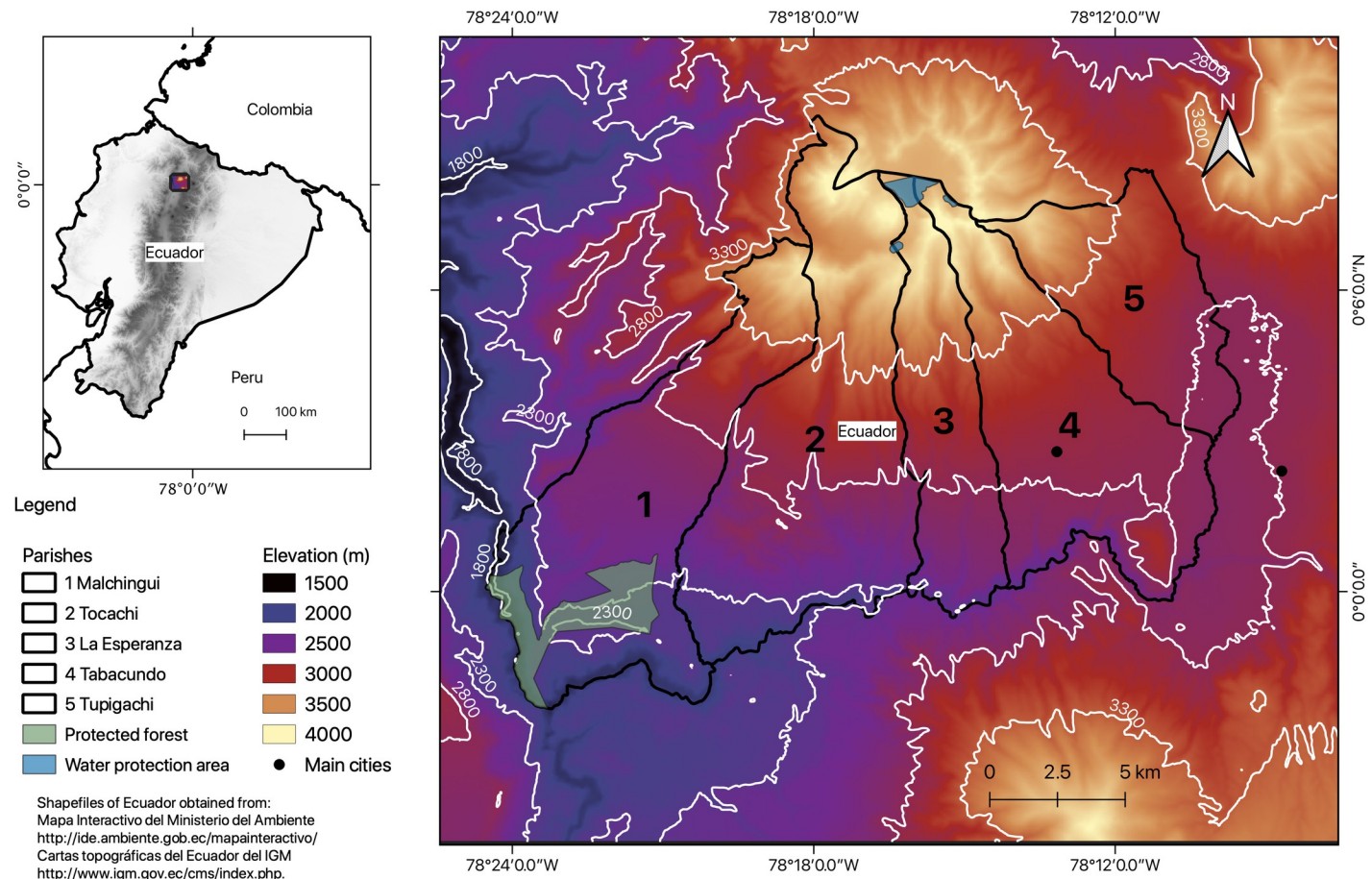

**Fig 2. Study area of the Pedro Moncayo county in the highlands of northern Ecuador.** (Data sources: ASTER Global Digital Elevation Model courtesy of NASA Earth Data. Made with Natural Earth. Free vector and raster map data @ naturalearthdata.com).

## Study area

Pedro Moncayo county is located in the western Andes of northern Ecuador (Fig 2). Pedro Moncayo is characterized by a wide elevation gradient (2400–4400 m) and a management regime that varies in intensity depending on the elevation [34]. The higher altitudinal zone (above 3300 m) is dominated by native ecosystems, represented by páramo and highland montane forests [33]. The middle altitudinal area (2800–3300 m) has been extensively used for agriculture and livestock through time, causing severe soil degradation [33, 35], and the lower lands are characterized by shrub dominated dry ecosystems (Fig 2).

The studied territory has a total surface area of 339 km$^2$, which is divided into 5 parishes that have an east to west geographical arrangement, depicting the same elevation belts previously described (Fig 2). Each parish shows different levels of production development and population trends. For example, parishes located to the west portray a local economy based on subsistence agriculture and lower population growth, whereas the eastern parishes are attracting a growing population, have a more concentrated urban development, more irrigation systems, and harbor an expanding agro-industrial sector [34].

Pedro Moncayo county is characterized by a typical climate of the tropical Andean region, with low annual variability but significant changes between night and day [36]. For example, quarterly midday maximum temperatures could range from 14°C to 24°C and minimum

night-time temperatures could range from 4˚C to 17˚C [36]. In contrast, the precipitation pattern follows a bimodal peak of heavy rains concentrated from October to November and April to May, followed by a dry period of low precipitation from June to September; quarterly precipitation could range from 0 mm to 225 mm, and depending on the season the territory could shift to a different hydrological regime [36]. For instance, from April to June the majority of the territory could have more than 200 mm of precipitation, whereas in the quarter of July to September most of the area receives less than 75 mm of precipitation [36].

Approximately 4% of the county's territory is designated as conservation or environmental management area, including the Jerusalem Protected Forest which occupies 1110 hectares of dry ecosystems in the county's lowlands, and the Mojanda Lacustric complex, protecting only 26 hectares of highland ecosystems and water sources (Fig 2) [34].

Although at present the majority (58.1%) of the territory of Pedro Moncayo is dedicated to traditional agricultural activities–mainly growing cereals, maize and potatoes–the economy of the region is based on the production and export of flowers (mainly roses) using greenhouse infrastructures [34]; small and medium-scale agriculture and livestock ranching are lower in terms of labor absorption, technology incorporation, and productivity [33].

## LULC datasets

To study landscape change through time in the study area, we used the official and publicly available LULC maps for four periods of time: 1990, 2000, 2008, and 2014 (http://ide. ambiente.gob.ec/mapainteractivo/). These are vector data produced by the Ministry of Environment (MAE) and the Ministry of Agriculture, Livestock, Aquaculture and Fisheries of Ecuador (MAGAP) in a mapping scale of 1:100,000 from mainly Landsat images (TM, 30 m). To obtain the LULC maps, a supervised classification method was carried out by a team of interpreters from MAE and MAGAP with training data of regions of interest (ROIs), using the maximum likelihood clustering algorithm of ENVI software [37]. The following overall accuracy values were obtained: 69%, 73%, 76%, and 85% for the years 1990, 2000, 2008, and 2014, respectively [38]. More details on the processing and classification methods used by MAE and MAGAP can be found here [37].

The LULC official classification encompasses a 2-level hierarchical scheme, based on the International Panel on Climate Change (IPCC) classes in combination with a taxonomy agreed by the entities in charge of generating land cover information in Ecuador [37].

To ensure the quality of the MAE-MAGAP LULC data sources for the study area, a process of validating the official vector maps from the different study periods was carried out. To support this validation process, as proposed by [8], distinct secondary sources of information were revised such as field points, Google Earth images, orthophotographs, and other official sources such as ecosystem coverage (http://ide.ambiente.gob.ec/mapainteractivo/), floriculture cadastral surveys, and other maps from the Ministry of Agriculture (http://geoportal. agricultura.gob.ec/). In addition, composite LANDSAT images from our study area, using radiometric enhancements and spectral band combinations, were also used [39]. From the validation process, five main typologies were improved [40]. These included: planted forests, developed areas (populated zones), floriculture (areas represented by greenhouses), and natural water bodies. Following the methods proposed by [41], a point-based accuracy assessment was conducted using Google Earth as a verification source. After that, a confusion matrix was created using 600 random points obtained from a stratified sampling scheme over the altitudinal bands. The resulting overall accuracy of the edited maps ranged from 82 to 86%. The validation process using visual digitalization over the LULC official vector layers from the periods of interest and the accuracy assessment were conducted in QGIS 3.10 [42].

For our LULC change analysis we used a modified categorization from MAE-MAGAP [37], we combined level 1 and 2 official LULC taxonomy (S1 Table). Briefly, we aggregated all the agricultural level 2 typologies into agricultural land, and as suggested by MAE-MAGAP [37] we included pasture in this LULC class since in the highlands of Ecuador there is a system of rotation from pasture to agricultural fields along the cropping cycles. In addition, we added floriculture crop as a separate typology from the developed LULC category, assuming that all greenhouses detected in the study region correspond to flower production based on the following facts: (1) The study area corresponds to the major center of floriculture production in the highland belt of Ecuador (above 2400 m), characterized by the implementation of greenhouse and irrigation technology mainly developed for the export market [43, 44]; (2) according to the flower export cadastral [45], the region of study encompasses thousands of greenhouses dedicated to flower production, occupying more than 1000 ha, and (3) the agricultural land in this region is characterized by a small-scale low input production system [34]. As a result, the identified LULC classes were 1) developed, 2) floriculture crop, 3) agricultural land, 4) planted forest, 5) shrubland and herbs, 6) native forest, 7) páramo, and 8) water bodies (S1 Table).

## Land use and cover changes

First, we mapped and estimated the land area occupied by each LULC class through time and the percentage change (C %) in each land-use class was calculated by dividing the area difference between the latest and the base year of each class by the coverage area in the base year and multiplying by 100 [8].

Then, LULC changes were estimated for three periods of analysis: 1990–2000 (T1), 2000–2008 (T2), and 2008–2014 (T3). To analyze the succession of LULC classes in these periods of analysis, we used discrete-time, finite-state, homogeneous (stationary) Markov chain models, which have been widely used to model LULC changes [46–48]. The Markov chain probability Matrix was estimated using the markovchain R-package [49] for five administrative zones (at the parish level) and across four elevation bands. By applying a Markov chain model for three periods of analysis to land use classes, it is possible to observe conversions between them when values are higher than 0.5. In contrast, the stability probability is observed when higher values are compared between the same LULC class, representing the probability of remaining in the same class in the consequent time period, given the present state of the class.

The spatial patterns of LULC change across administrative zones were obtained from an overlay procedure of the LULC maps with the polygons of parishes from the studied Pedro Moncayo county, which were downloaded from the official reference (https://www.ecuadorencifras.gob.ec/clasificador-geografico-estadistico-dpa/) [50]. In the same way, to understand the patterns of LULC change across elevation classes, first the Global Digital Elevation Model (ASTER GDEM) at a 30 m spatial resolution was downloaded from NASA's Earth Data website, was clipped to the study area and the resulting image was further reclassified according to elevation bands, with an interval of 500 m as proposed by [41, 51]. Then, the following four elevation bands <2300, 2300–2800, 2800–3300, >3300 m (Fig 2) were obtained for the study region. Finally, the LULC classification for each year was layered over both (1) the reclassified elevation map, and the (2) reclassified administrative map. Spatial data assimilation, processing, and overlaying analysis were conducted in R [52].

## Drivers of change

To understand which driving forces could explain LULC transitions in our study region, we selected a set of factors within our DPSIR framework (Fig 1) if they meet the criteria selection exactly as proposed by [53]: '(1) Relevancy: indicators should reflect the underlying cause of

**Table 1. Direct driving forces are included as predictors in the generalized additive model to explain probability of change of LULC transitions.**

| Type | Name | Units | Description | Spatial resolution | Source |
|------|------|-------|-------------|--------------------|--------|
| Socio-economic driving forces | Education index | N/A | Change of a compounded index of eight census indicators of education, with parish breakdown, between years | Census areas | Instituto Nacional de Estadísticas y Censos [57] (1990 & 2001, 2010, 2014*) |
| | Index of economic diversification of employment | N/A | Change of index of economic concentration of employment between years of study | Census areas | Instituto Nacional de Estadísticas y Censos [57] (1990 & 2001, 2010) |
| Demographic and infrastructure variables | Total population | Number of inhabitants | Change of total population between years of study | Census areas | Instituto Nacional de Estadísticas y Censos [57] (1990 & 2001, 2010) |
| | Distance to roads Distance to nearest cities | km | Change of distance to roads or nearest cities between years of study | 30 m | Cartography–Instituto Geográfico Militar and digitation from Landsat images (1990, 2000, 2008) |
| Climate factors | Maximum temperature | ˚C | Change of daily maximum air temperatures at 2 meters averaged over each month and summarized in a year | 1 km | Chelsa datasets tmax, tmin, prec (1990, 2000, 2008) [58] |
| | Minimum temperature | | | | |
| | Precipitation | mm | Change of monthly means of daily forecast accumulations of total precipitation at earth surface summarized in a year | | |
| | Water availability by irrigation | N/A | Availability of water from the main irrigation system | 30 m | Digitation from Google images (2008) |
| Topographic | Altitude | m | Height in relation to sea level | 30 m | ASTER Global Digital Elevation Model courtesy of NASA Earth Data |
| | Aspect | degrees | Orientation of slope, measured clockwise in degrees from 0 to 360 | | |
| | Slope | % | Steepness or the degree of incline of a surface | | |
| Governance decisions on production development | Parish typologies | N/A | Gradient of production development (1–5) based on policy decisions across administrative zones | Parish | Land use development plan of Pedro Moncayo county (2015) [34] |

environmental change. (2) Availability: the indicator data should be available, accessible, and consistent within the period of analysis. (3) Independence: indicators must be independent of each other to eliminate multicollinearity. (4) Representativeness: each indicator used in the model must represent a category or phenomenon of its own and must provide superior information to other indicators in a similar category'.

Criteria 1 was achieved by conducting a literature review to select a list of driving forces that have been documented to explain LULC change in tropical mountain systems [6–8, 10, 14, 15]. Data availability was the result of searching freely available and accessible databases both from national and international sources for the period of interest (Table 1). To meet criteria 3 and 4, we selected groups of drivers that represent different complementary phenomena to explain LULC changes (Table 1). We avoided multicollinearity within each group of drivers by conducting a principal component analysis (PCA) to discard highly correlated variables.

The result was a compiled dataset of 13 variables considered to be direct drivers, organized into the following groups: (1) socio-economic, (2) demographic and infrastructure factors, (3) topographic and (4) climate variables, in addition to (6) local governance decisions about landscape development that influence landscape transitions (Table 1, Fig 1).

In order to increase the number of units of analysis within parishes, all these variables were obtained at the spatial resolution of census area [54]. After the spatial data assimilation, processing and visualization necessary to obtain the drivers at the spatial unit of analysis, we carried out a reduction dimension procedure using PCA [55] for each grouping of drivers. Within the PCA, correlated variables were screened for the total variation explained by the

first principal axes, and used to remove correlated variables [56]. Coordinates of the principal components that accounted for more than 60% of the variation were then used as explanatory variables in a subsequent statistical model to reduce the dimensions of the multivariate matrix within each grouping of drivers.

## Statistical analysis

We synthesized and incorporated the different groupings of drivers into a statistical model to improve LULC predictions and inform decision making by carrying out multivariate analysis using Generalized Additive Models (GAM). GAMs are an approach used extensively in environmental modeling, and provide great scope to model complex relationships between covariates [59, 60]. We used GAM regressions to elucidate two types of transitions in our study area: 1) the probability of natural ecosystem loss, and 2) the probability of change to anthropic environments. The LULC trends evaluated as response variables within the first approach (Fig 1) were the probability of loss of native forest, páramo, and shrubs and herbs estimated through Markov chain analysis. Complementarily, the second approach tried to explain what drivers could cause the transitions towards developed areas, floriculture crops, and food crop and pastures (Fig 1). We did not include transitions to planted forests because this LULC element was shown to be very stable during the periods of analysis. As explained in the previous section, the explanatory variables for each GAM were the coordinates of the PCAs that explained more than 60% of the variation in the multivariate matrix of each driver grouping.

The computational methods for the GAM modeling were implemented from the Comprehensive R Archive Network (CRAN) repository 'mgcv' package [61]. Since in our study, the response variable is a probability ranging from 0 to 1, we used the beta regression within the GAM family, as suggested by this type of data [62]. For the smoothing basis function, we used the penalized cubic regression spline to lower computation cost and avoid overfitting; the smoothing parameter estimation was restricted maximum likelihood ('REML'), typically used for smooth components viewed as random effects [59]. After checking the results of different models using distinct methods for selecting the number of knots (default, cross validation, and manual adjustments), we selected the more conservative approach. We set the number of knots to three to be flexible enough to allow the models to fit simple curve relationships, preventing spline curves with complex overfitting estimates. Overfitting curves would have limited our ability to interpret and describe the mechanisms operating, in order to explain LULC changes from an ecological perspective. We presented the results of the GAMs with Partial Dependence Plots using the 'mgcv' R-package [61] to determine which variables best explained the variation in LULC change [59].

## Results

### Coverage area patterns and land-change dynamics through time

Agricultural land was the most representative LULC type in the study area, followed by shrubs and herbs (Fig 3). Both LULC types were very dynamic over the different periods of analysis: agricultural land ranged from 35 to 50% of the total area, and shrubs and herbs varied from 16 to 28% of the total area, depending on the period analyzed (Table 2).

Overall, natural ecosystems–which are mainly represented by native forests and páramos– decreased from 1990 to 2014 (Table 2), there was a 40% and 16% decrease of native forest and páramo cover when comparing the first and last periods of study (Table 2); but, areas of páramo still represent an important part (13%) of the study territory in the last period of study. Natural water bodies (lakes and rivers) showed high persistence over time (Table 2).

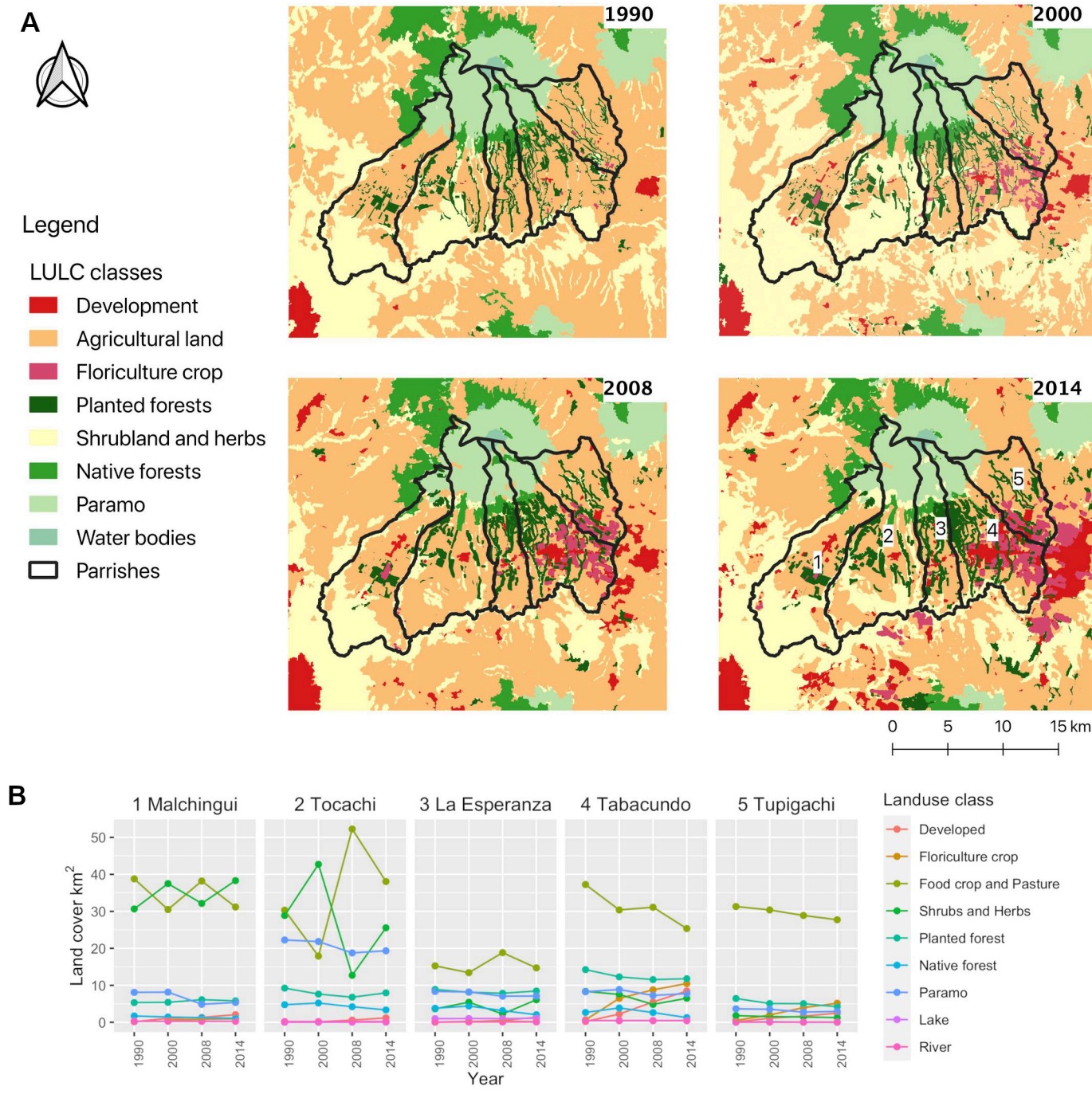

**Fig 3. Land use land cover changes in Pedro Moncayo county through time.** A. LULC maps throughout the periods of study (1990, 2000, 2008 and 2014). B. Land extent changes through time in Pedro Moncayo county by administrative zones (parishes).

Developed areas and floriculture crops continuously increased over time, and although they were poorly represented in the first period of analysis (less than 0.4% in 1990), by 2014 they represented almost 5% of the study area (Table 2), demonstrating a 26 and 13-fold increase from 1990 to 2014, respectively.

**Table 2. Changes in land cover classification in Pedro Moncayo county from 1990 to 2014.**

| LULC TYPE | YEAR | 1990 | | 2000 | | 2008 | | 2014 | | 2014–1990 |
|---|---|---|---|---|---|---|---|---|---|---|
| | class | km² | % | km² | % | km² | % | km² | % | % Change |
| Developed | | 0.58 | 0.17 | 4.72 | 1.39 | 9.61 | 2.84 | 15.54 | 4.60 | 2569.55 |
| Floriculture crop | | 1.19 | 0.35 | 9.44 | 2.79 | 14.06 | 4.16 | 16.75 | 4.95 | 1305.89 |
| Food crop and pasture | | 152.92 | 45.20 | 122.58 | 36.23 | 169.29 | 50.04 | 137.03 | 40.51 | -10.39 |
| Planted forest | | 44.16 | 13.05 | 38.56 | 11.40 | 37.35 | 11.04 | 38.24 | 11.30 | -13.42 |
| Shrubs and herbs | | 73.33 | 21.68 | 94.74 | 28.00 | 53.40 | 15.78 | 77.75 | 22.98 | 6.02 |
| Native forest | | 12.96 | 3.83 | 15.11 | 4.47 | 11.29 | 3.34 | 7.77 | 2.30 | -40.03 |
| Páramo | | 50.62 | 14.96 | 50.60 | 14.96 | 40.73 | 12.04 | 42.51 | 12.57 | -16.03 |
| Lake | | 1.48 | 0.44 | 1.52 | 0.45 | 1.52 | 0.45 | 1.66 | 0.49 | 12.29 |
| River | | 1.04 | 0.31 | 1.06 | 0.31 | 1.04 | 0.31 | 1.04 | 0.31 | 0.09 |
| Total | | 338.29 | 100.00 | 338.33 | 100.00 | 338.29 | 100.00 | 338.29 | 100 | 0.00 |

Landscape dynamics through time were not homogenous across the study area, instead they show a geographic pattern (Fig 3). Expansion of developed areas and floriculture crops occurred mainly in the southeastern part of the studied region (Fig 3). The greatest degree of loss of native forests and páramos occurred in the northeast (Fig 3), where there is almost no páramo left due to the expansion of agricultural land.

## Transitions of native ecosystems

In general, as expected, the stability of native forests is decreasing through time across the entire territory (Fig 4), with the exception of the western parish where the probability of remaining in this LULC class increases through time–probably due to agricultural land abandonment (Fig 4). In contrast, areas located in the east tend to have lower values of stability through time and higher probabilities of changing to páramo and agricultural land; this pattern was more evident in the last period evaluated (2008–2014) (Fig 5). Additionally, this trend is more evident along elevation bands; where native forests located above 3300 m showed a lower probability of remaining as forest through the years (Fig 5) and in the 2800–3300 m altitudinal belt there is a high probability of converting native to planted forests, especially in the center of the territory (Fig 5).

Furthermore, shrubs and herbs show variable change throughout the study period (Fig 4).

In the majority of administrative areas, the stability of shrubs and herbs decreased (from values around 0.75 to values close to 0.25) in the second period of evaluation (2000–2008) and increased again in the last period (2008–2014). Across all elevation belts, this LULC class tended to follow a dynamic trend changing back and forth with the agricultural land; however, this pattern was not observed in the eastern parish at the elevation belt of 2800–3300 m, where the landscape seems to have a high probability of remaining as agricultural land (S1 Fig).

In contrast, páramo is the most stable among all the natural ecosystems evaluated, although a slight decrease in stability was observed from values above 0.90 to around 0.75 in the second period of analysis (2000–2008) (Fig 4), and the probability of remaining in the same land use class increased by the last period of analysis (2008–2014). Since this ecosystem is characteristic of highlands (above 3000 m) the transition probabilities were only observed for the two higher elevation belts evaluated and their stability seems to be increasing in the administrative zone located in the western part of the territory (S2 Fig).

## Transitions to anthropic environments

Developed areas demonstrate a differential trend over time in the study area (Fig 6). In the western areas of the territory (Fig 6) the stability of this LULC class decreased in the second

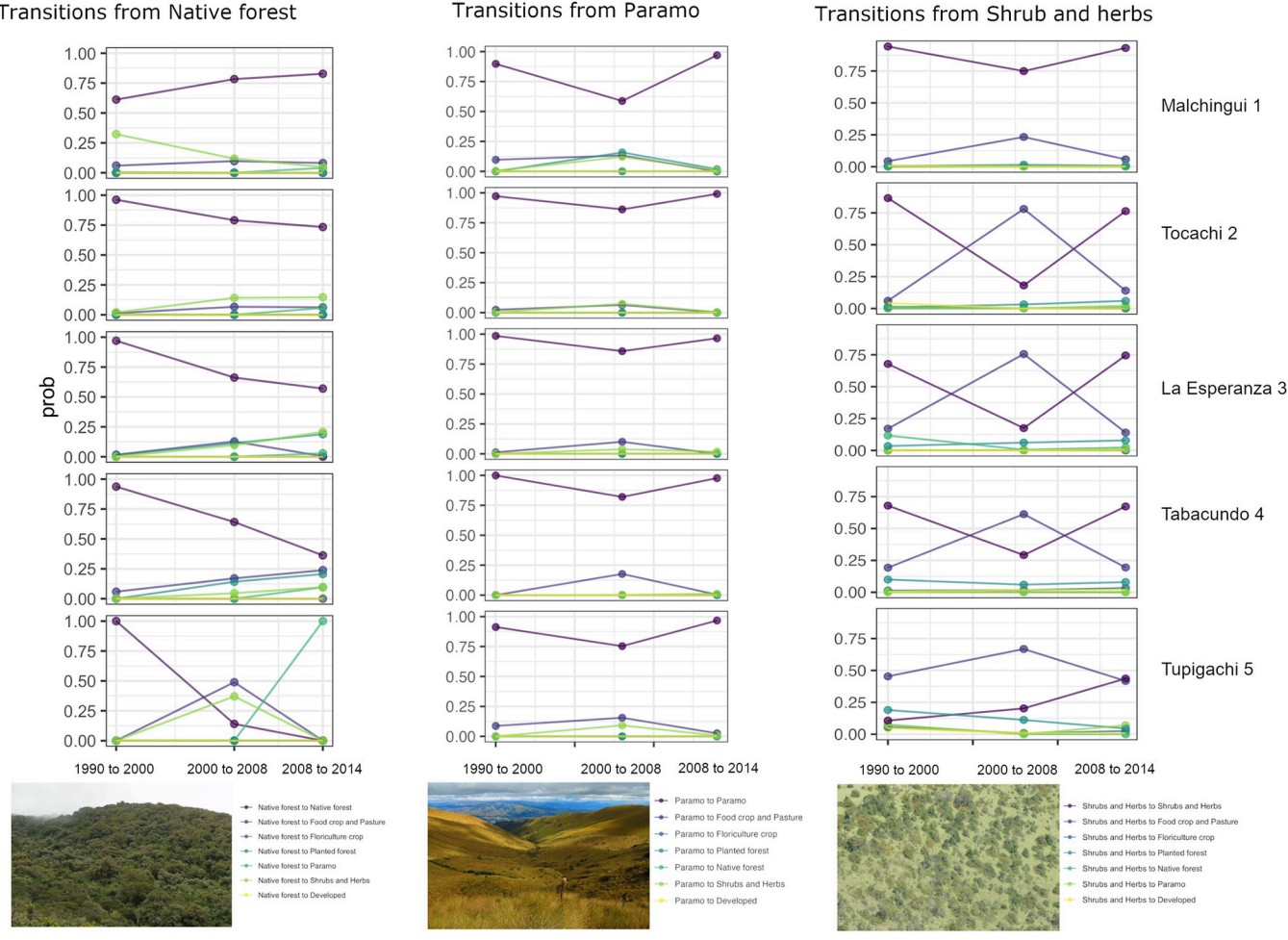

**Fig 4. Transition probability of native ecosystems through time in Pedro Moncayo county, at the parish level.** (The above photos are the original works of the authors).

period of evaluation (2000–2008) and significantly increased again in the last time period (2008–2014). In contrast, the parishes located to the east exhibit a more stable probability of remaining as developed areas through time, probably due to their proximity to the larger towns (Fig 6). Since the territory studied is in general a rural area, there is a dynamic trend towards converting agricultural land to urban areas, which follows a geographic pattern (Fig 6).

To the east and center of the study area, floriculture crops have not been fully established because land use tends to change to agricultural land (Fig 6); in contrast, this LULC type located in the eastern parishes is more stable with values around 0.75 throughout the period of study (Fig 6).

Agricultural land is a very stable land use class throughout the study period in the administrative zones located in the center and eastern parts of the study area, with values ranging above 0.77 (Fig 6); the stability of this land use class in the west followed a dynamic trend through time: in the first period (1990–2000) it was lower than in the second period of analysis, and it increased again by the last period studied. In contrast, planted forests depict a very stable land use trend through time across the territory, their probability of remaining in the same land use class ranges from 0.6 to 0.90 (Fig 6).

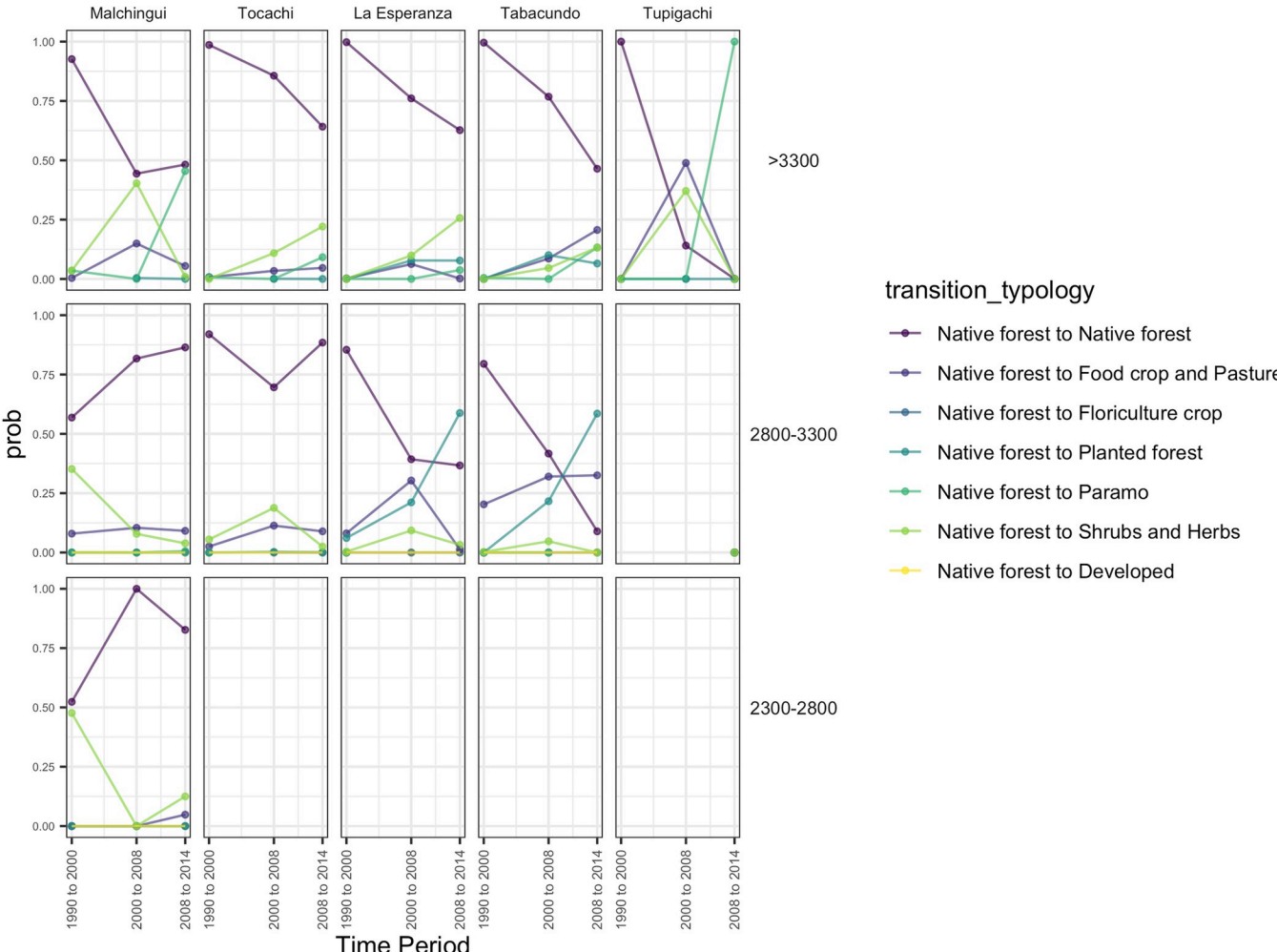

**Fig 5. Transition probability of native forests through time in Pedro Moncayo county, by altitudinal bands at the parish level.**

## Drivers of change

The general results of the screening and dimension reduction process within each driver grouping obtained after conducting PCAs are briefly described as follows: within the socioeconomic drivers group, variables were not correlated and the first PC explained much of the variability (60%) in this matrix. Inside the topographic group, aspect was removed after data screening and the PC1 coordinates were selected for the further models, since they accounted for 61% of the variation. For instance, tmax was a correlated variable and it was extracted from the climate matrix, then both PC1 (representing irrigation and tmin variability) and PC2 (representing water availability by irrigation) were selected as predictor variables, as they together accounted for more than the 60% of the variation in the climate dataset. Finally, in the driver grouping that represents demography and infrastructure, distance to roads was removed and the coordinates from PC1 (distance to cities) and PC2 (population change) were included as predictors within this driver grouping because together they explained more than 60% of the variation.

The selected predictors or possible drivers of change to explain LULC transitions displayed different spatial distributions within the study area (S3 and S4 Figs), depicting a territory with

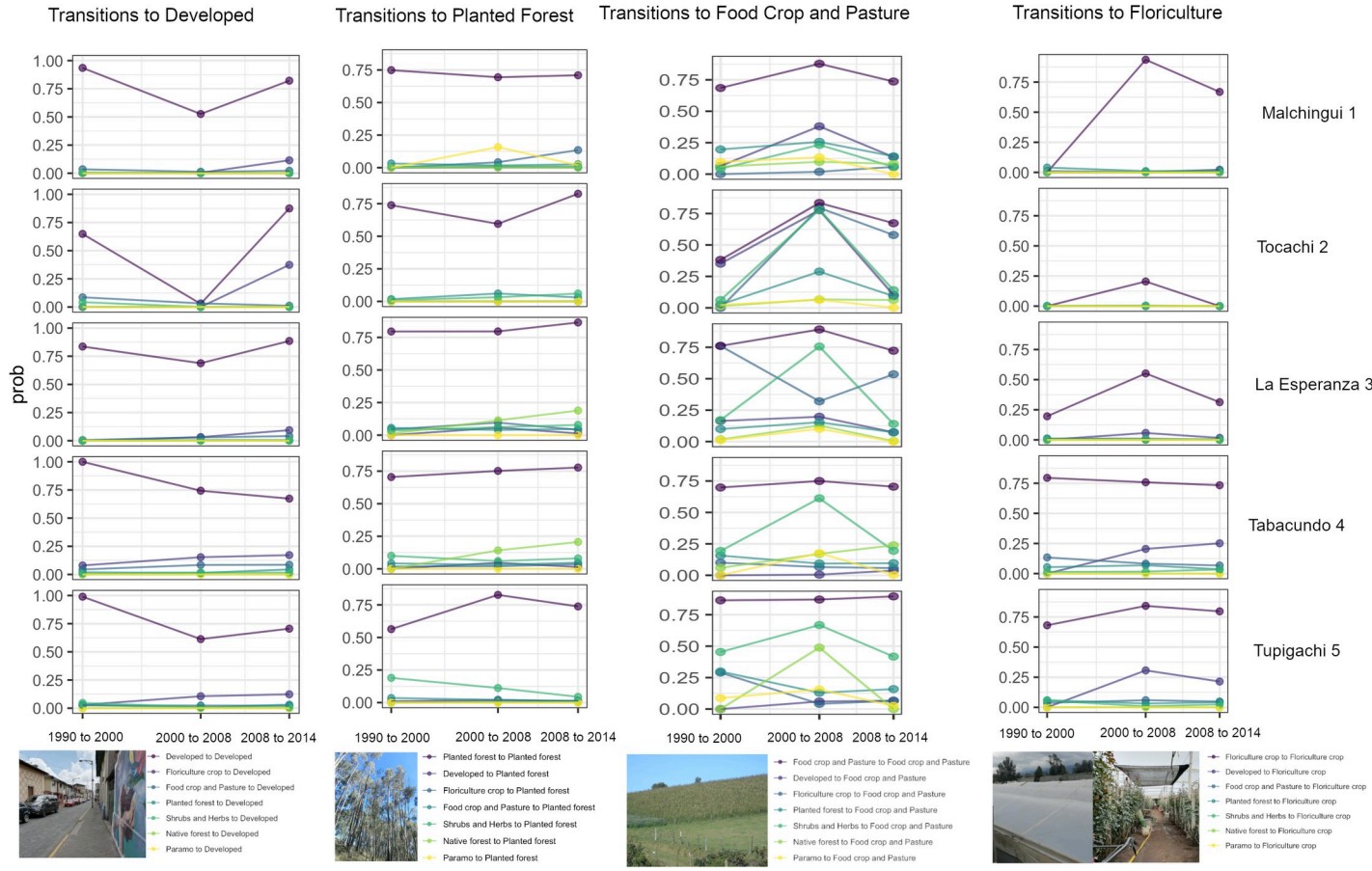

**Fig 6. Probability of land use transition to anthropic environments in Pedro Moncayo county, at the parish level through time.** The above photos depicting the developed area and the floriculture crops are reprinted from Guasgua (https://zenodo.org/record/6231701#.YhVzVpPMLJ8) under a CC BY license, with permission from (Jessica Guasgua), original copyright 2022. The other photos are the original works of the authors.

contrasting patterns. The details of the spatio-temporal distribution of the drivers of change are presented in S3 and S4 Figs.

Table 3 describes the results of the different LULC transitions studied and their main explanatory variables; the GAMs demonstrated different results when explaining each LULC transition (Table 3). The lowest total variance (21.00%) corresponded to the native forest loss model and the largest value (41.80%) was for the agricultural expansion model. Overall, the most relevant parameters explaining LULC in the region were the topographic driver grouping (which incorporates elevation and slope), this driver grouping was highly significant for the majority of the transitions studied (p<0.001, Table 3), with the exception of the shrub and herb loss. In contrast, the climate driver grouping PC1 (which mostly depicts the variation of precipitation and minimum temperature) was not significant in any model (p>0.05).

For the native forest loss model, the most important groupings of drivers (p<0.001) were the socioeconomic and topographic drivers (Fig 7, Table 3). For instance, páramo loss was only explained by the variation in elevation and slope (topographic PC1) (Table 3, S5 Fig). Fig 7 shows the GAM partial dependence plots for the native forest loss model and indicates that the probability of native forest loss increases as land aspect PC1 increases, in other words, when elevation and slope increases. In contrast, when the socioeconomic variables have low

**Table 3. Summary of the results of the generalized additive models to elucidate drivers of change for the six LULC transition models in Pedro Moncayo county.**

| Drivers | Native forest loss | | | Paramo loss | | | Shrub loss | | | Urbanization | | | Floriculture expansion | | | Agriculture expansion | | |
|---|---|---|---|---|---|---|---|---|---|---|---|---|---|---|---|---|---|---|
| | p value | chi sq | edf | p value | chi sq | edf | p value | chi sq | edf | p value | chi sq | edf | p value | chi sq | edf | p value | chi sq | edf |
| **Socioeconomic PC1** | **0,000** | 19,65 | 1,8 | 0,721 | 0,00 | < 1 | 1,000 | 0,00 | < 1 | 0,468 | 0,00 | < 1 | 0,307 | 0,05 | < 1 | 0,812 | 0,00 | < 1 |
| **Topographic PC1** | **0,000** | 13,32 | 1,4 | **0,000** | 23,89 | 1,5 | 0,426 | 0,00 | < 1 | **0,000** | 16,69 | 1,7 | **0,001** | 9,62 | 1,4 | **0,000** | 18,48 | 1,9 |
| **Climate factors PC1** | 0,889 | 0,00 | 1,0 | 1,000 | 0,00 | < 1 | 1,000 | 0,00 | < 1 | 1,000 | 0,00 | < 1 | 0,774 | 0,00 | < 1 | 0,814 | 0,00 | < 1 |
| **Climate factors PC2** | 0,336 | 0,00 | < 1 | 0,791 | 0,00 | < 1 | 0,374 | 0,00 | < 1 | 0,609 | 0,00 | < 1 | **0,018** | 4,23 | < 1 | 0,039 | 3,06 | < 1 |
| **Demography & infrastructure PC1** | 0,547 | 0,00 | < 1 | 1,000 | 0,00 | < 1 | **0,000** | 17,02 | 1,76 | **0,000** | 26,71 | 1,8 | 0,936 | 0,00 | < 1 | **0,000** | 15,21 | 1,3 |
| **Demography & infrastructure PC2** | 0,189 | 0,72 | < 1 | 1,000 | 0,00 | < 1 | 0,364 | 0,00 | < 1 | **0,048** | 0,00 | 1,2 | 0,144 | 1,24 | < 1 | 1,000 | 0,00 | < 1 |
| **Parish governance** | 0,507 | 0,00 | < 1 | 0,118 | 1,46 | < 1 | **0,000** | 25,53 | 1,53 | 0,386 | 0,00 | < 1 | **0,000** | 10,41 | 1,2 | **0,000** | 31,00 | 1,8 |
| **Deviance explained** | 21,00% | | | 20,80% | | | 39,90% | | | 36,50% | | | 25,00% | | | 41,80% | | |
| **R-sq.(adj)** | 0,22 | | | 0,15 | | | 0,29 | | | 0,30 | | | 0,21 | | | 0,33 | | |

and high values the probability of forest loss increases, although the confidence interval for lower values in the socioeconomic drivers is higher.

Some transition models were explained by similar driver groupings, such as shrub loss and agricultural expansion (Table 3). These also show a contrasting pattern in their response variables, in such a way that when an increase in agricultural areas was prevalent, there was a decrease in shrub and herb land (Fig 2). These models depicted the following driver groupings as significant parameters (p<0.001): pressure drivers (PC1) and a variable that describes differences in the development of distinct administrative areas within the study area (Table 3).

S6 and S7 Figs show the GAM partial dependence plots for shrub and herb loss and agricultural expansion, respectively, and they reveal that high probabilities for these transitions are related to medium values of elevation and slope (topographic drivers). Additionally, when the main local cities are further away the probability of converting natural areas to agricultural land increases, and there is a linear increase in rates of change to agricultural land with the gradient of development at the parish level.

Variables leading to the highest change in the probability of transition to floriculture crops comprise the topographic driver grouping, the climate PC2, which includes water irrigation and the development gradient across parishes. Floriculture crops increase as elevation and slopes decrease. Complimentarily, when more water is available through irrigation, the probability of establishing floriculture crops increases (S8 Fig).

The urbanization transition model was explained by topographic, demographic, and infrastructure driver grouping PC1 (p<0.001) (Table 3). Urban transition probabilities decrease significantly (p<0.001) with altitude and slope, it also significantly decreases (p<0.001) with the distance to city centers (demographic and infrastructure PC1), and with higher values of total population change (demographic and infrastructure PC2) (S9 Fig).

## Discussion

This study demonstrates that a combination of environmental variables and human induced factors still have an impact on LULC transformations during the past several decades, despite a legacy of landscape transformation occurring in the Ecuadorian highlands [12, 63] and

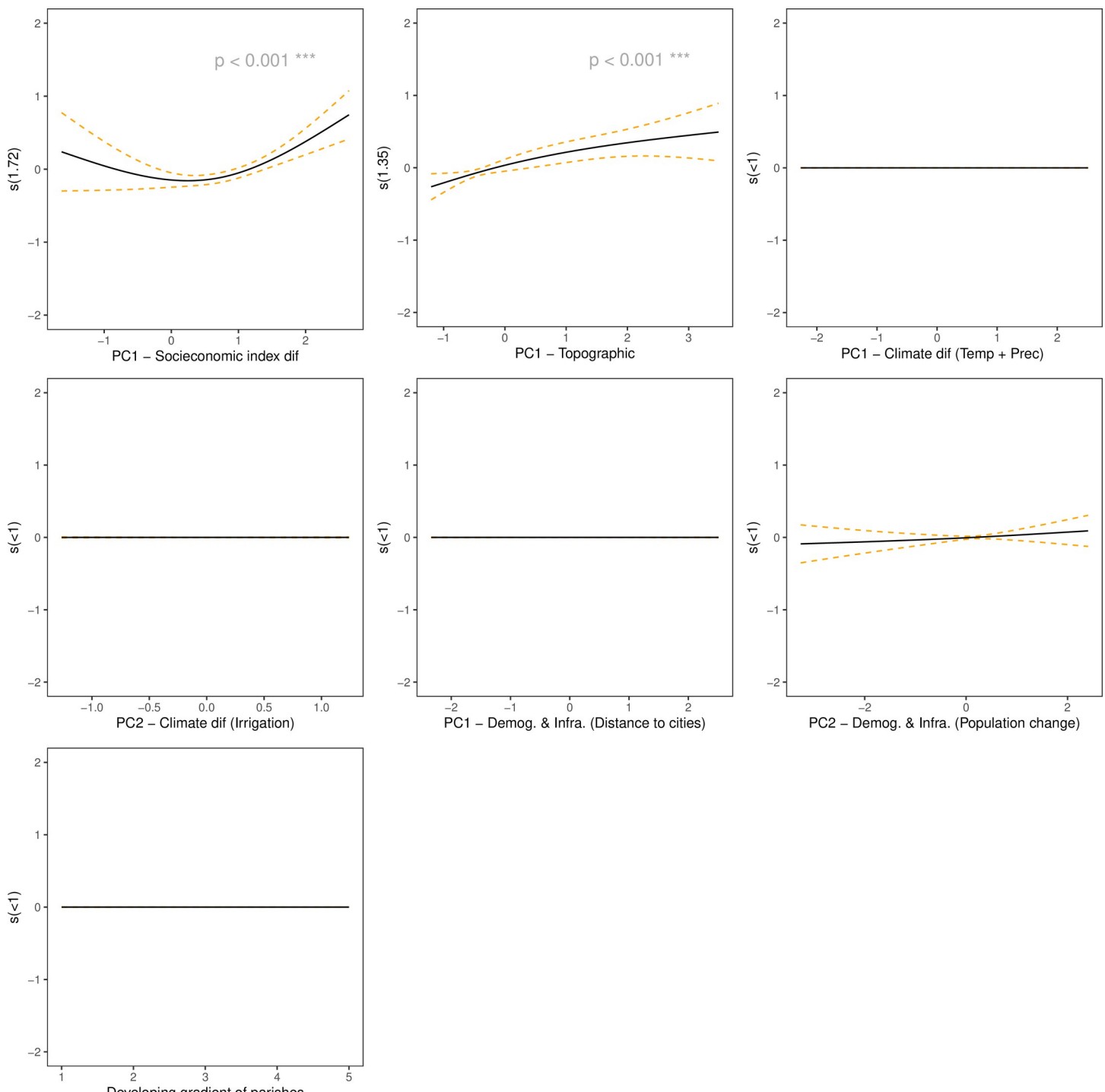

**Fig 7. Generalized additive model partial dependence plots for native forest loss.** Each plot shows a covariate and their partial dependence on probability of native forest loss in the context of the model. The y axis shows the mean of the probability of native forest loss and the x axis the covariate interval. The gray area represents the 95% confidence interval.

supports findings in similar mountainous landscapes of Latin America [6, 8–10]. The study area of Pedro Moncayo represents a rural Andean landscape dominated by an agricultural matrix which contains important areas of shrubland and páramo, accompanied by patches of

remaining native forest, consistent with other current landscapes in the Tropical Andes [6, 8, 10, 12].

We found spatially explicit patterns of LULC transition across the study area, including a distinct deforestation pattern of native montane forests located below 3300 m.a.s.l. In addition, we found an unexpectedly high pattern of páramo stability for the majority of the studied territory, and a dynamic transition between agricultural land and shrubland. Likewise, we found an exponential increase in urban land and floriculture crops in the eastern part of the territory. This result is striking because of the small spatial scale where the changes occur; our study area encompasses only 334 km$^2$ compared to other landscapes studied in central Ecuador [12], the Peruvian Puna [8] or Colombia [10], where the extent of land is 10, 120, and 800 times larger, respectively than our studied territory.

We estimated a páramo loss of 16% from 1990 to 2014 in Pedro Moncayo county (Table 1), this result is consistent with the findings of loss (13%) in a nearby territory [64]. Although most of the studied territory depicted a relatively high pattern of páramo stability, as also described for a highland landscape of central Ecuador [12], our results also demonstrated a hotspot of páramo conversion to agricultural land concentrated in the northeast. In contrast, our results are strikingly different to the land cover patterns observed in other páramos in the region, where a more widespread agricultural use of páramo was observed [65, 66]. Another common transition reported for páramos in the Ecuadorian mountains is to exotic timber plantations [16], yet this trend was not apparent for our studied territory.

We found a 40% montane forest loss from 1990 to 2014, and the Markov chain model demonstrated a very low probability of persistence of this ecosystem in the majority of Pedro Moncayo county (Figs 2 and 4). This is consistent with the general trend of deforestation and degradation of mountain forests in the Tropical Andes mainly explained by agricultural expansion [67]. We also found that the highest chances of loss occur in the altitudinal band of 2800 to 3300 m (Fig 3). These findings are in accordance with those described for other representative highlands in central Ecuador [12]; however, LULC change studies carried out in more isolated landscapes of central and southern Ecuador reported deforestation hotspots for lowland montane forests and afforestation transition in upper altitudinal areas [9, 13]; additionally, higher rates of deforestation were also observed in the lowland forest of Colombia, and in the Napo region along the northeastern Ecuadorean border [10].

Mountain forests are considered one of the most threatened forest types in the tropics [68], which are also highlighted as a global priority for conservation due to their relatively high biodiversity and high level of endemism [69], and their vital role in the provision of different ecosystem services in the region [70, 71]. However, if the trends demonstrated by the Markov model are maintained for this territory, there is a high probability that the remnant montane forests will be permanently lost in a few years, posing a greater threat to the already vulnerable biodiversity [72] and limiting the capacity of these ecosystems to provide services in the county, such as the provision and regulation of freshwater, "wild foods", and many other non-timber forest products [20], as described for other latitudes [3, 73, 74].

Along with this deforestation trend, we observed a dynamic and opposite transition between agriculture areas and shrubland, this pattern was more evident for the parishes located in the center of Pedro Moncayo county and along the elevation bands between 2300 to 3300 m. This pattern could demonstrate a gain of secondary vegetation, probably due to a temporal abandonment of agricultural areas, followed by a net gain of agricultural land which has been observed in other Andean systems of Colombia [10] and Central America [75].

We found that urban areas are dramatically increasing in the eastern part of the territory (Fig 1, Table 1); we reported a 25-fold increase in urban cover from 1990 to 2014. This pattern follows the global trend of urban expansion [76, 77], but the rate of expansion is even faster

than that reported for many cities around the world [78] and in small urban centers [79], raising questions of the sustainability of future development in the region. For example, higher probabilities of urban land expansion were explained by increases in population, proximity to urban centers, and occurred at lower elevations and slopes in previous crop land. This pattern has been observed in other regions of South America, where urban expansion is taking place largely on agricultural land [76], a zone characterized by areas of lower altitude and slope, which in the Andean zones corresponds to the more fertile valleys between mountains.

Another interesting finding was the exponential expansion of flower cultivation cover reported for Pedro Moncayo county (Table 1, Fig 2). We described a 13-fold increase in total land area of greenhouse floriculture from 1990 to 2014 (Table 1); this expansion was observed primarily in the eastern parishes of the territory (Figs 1–3), which are located contiguous to Cayambe county, another center for the development of this activity in Ecuador [64]. This region, situated in Pichincha Province in central Ecuador, has an equatorial location and has optimal sunlight conditions (long hours of daylight) and an ideal highland climate (abundant sunshine, warm days and cool nights), making it possible to produce some of the highest quality flowers in the world [43, 44] and proximity to international airports and key infrastructure facilitates product export.

Our analysis suggests that in addition to the topographic variables, another driver that explains the floriculture expansion pattern is water availability by irrigation, depicted by the geographic pattern of irrigation in the lower eastern part of the studied territory. This creates a subsidy for growing crops which would have been limited by natural precipitation, as demonstrated by [80] to increase yield in many crops. This irrigation canal transports water from the glacier of a snow-capped mountain located in a contiguous territory, corresponding to the neighboring county (Cayambe). This water source only reaches the center of the territory and can distribute water to lower elevations, therefore providing a water irrigation subsidy to the area situated to south-east.

We found that topographic variables (elevation and slope) are the most important drivers for all LULC transitions. For instance, native ecosystem transitions (including the models to explain loss of native forest and páramo) and agricultural expansion were both significantly related to changes in elevation and slope, in such a way that the probability of native ecosystem loss and the probability of agriculture expansion increase with elevation and slope, until they reach a certain value where they level off (native forest and páramo models) and even decrease (shrub and herb loss and agricultural expansion models). These complementary trends suggest that the major pressure on native ecosystems in this region of northern Ecuador is the continued expansion upwards of the agricultural-livestock frontier, similar to other Andean landscapes [10, 12]. In addition, the expansion of urban areas and floriculture crops in the previous agricultural land, located at lower elevations of the eastern part of the territory, represents ongoing pressure for expansion of the agricultural frontier in highland areas. Even though we did not find evidence that climatic variation explained the LULC transitions, the effect of climate change could be stronger in the near future due to the extreme events predicted in the tropical Andes [21], affecting the capacity of highland ecosystems to keep providing key goods and services to people [81].

The trend of native ecosystem loss associated with higher elevation and slopes observed in this landscape of northern Ecuador could be attributed to its past patterns of land use, as summarized by [5, 82]. The most drastic transformation and loss of native ecosystems in Andean landscapes occurred centuries ago and this was also expanded in the mid-twenty century by agrarian reform; current native ecosystems are only the remnant patches, localized at higher elevations and slopes [20]. However, the leveling off and further decrease in the probability of native forest loss at higher values of topographic variables could be explained by conservation

measures adopted to restrict human activities in the upper mountain belt, such as the establishment of protected areas [5, 12, 20] or implementation of national or local policies to limit agricultural expansion [4] that have prevented the loss of high mountain ecosystems in other Andean regions [6, 17].

Páramos and other high-elevation ecosystems (pristine native forest patches), which are ecosystems situated above 3500 m in the northern highlands of Ecuador, are currently more valued due to their importance in providing critical ecosystem services and, thus, in Ecuador have received special protection measures at the national [83, 84] and local level [85].

Studies have found that environmental variables such as topography were better predictors of woody vegetation change, indicating that these variables place physical limits on the types of land-use practices that are feasible in a region [6, 8, 10]. However, the trends were different from those observed in our study, in that these authors found that deforestation occurred in the lowlands, which are more appropriate for large-scale mechanized agriculture [6, 8, 10].

The dynamic transition trend between agricultural land and shrubland observed in our study could be attributed to natural reforestation succession at high elevations (e.g., cooler temperatures, steeper slopes), which is consistent with other findings [6, 13]. In our study, this pattern was also associated with variation in population change, which could be attributed to population migration dynamics within the territory. Migrations of farmers from higher mountainous zones to urban concentrated areas have been widely documented in different regions of Latin America and are the drivers associated with natural reforestation in higher elevations due to agricultural land abandonment [17]. This finding is consistent with the local demography dynamics, where the urban population tripled from 1990 to 2010 (from 3,000 to 10,000 inhabitants) while the rural population has doubled (12,000 to 23,000 inhabitants) in the same period [34], representing an increasing pressure on natural resources to sustain livelihoods in the region.

In places where this landscape transition has been reported, it has facilitated ecosystem recovery in the highlands, likewise this has allowed the provision of ecosystem services to be maintained for a growing urban population [17]. The dynamic conversion from agricultural land to shrubland in some highland areas of this landscape, explained by rural-urban migration, is consistent with the "Forest Transition Model" proposed by Mather [86]. In our study area the pattern was uneven; for instance, native forests are decreasing in some areas, while shrubland was expanding in other areas, describing a process of ecological succession before a fully recovered forest could occur. Maintaining and increasing native ecosystems in higher elevations and expanding urban and agricultural areas in the lowland and valleys raises new opportunities and challenges for conservation. However, the consequences of these spatial transitions have not been studied in depth [17].

We have considered a comprehensive set of factors characterizing landscape conversion dynamics, however some limitations concerning the scope of the drivers used for this analysis should be considered. The underlying driving forces affecting land use transformations could also be attributed to production support policies geared towards the internal market and exports [12, 14], which were not included in our analysis. For example, the greenhouse floriculture expansion initiated in the 1990s has been cited as a response to favorable trade agreements and increased access to technologies from multiple sources and local entrepreneurship [44]. Flower cultivation is a land- and labor-intensive activity with high land productivity (that is, high market value of output per hectare) [43]. However, the gains in income have surely been offset by growing health and environmental problems posed by the intensive use of pesticides in flower cultivation [43] and irreversible change to landscape properties.

All indications suggest that flower exports will continue to play a major and probably increasing role in Ecuador's economy [43]; in fact, this industry is steadily expanding and

causing land use changes in the territory; for instance, former important and traditional lands dedicated to livestock and food crop production, located in areas with the capacity for agricultural production and with access to irrigation systems have been transformed into greenhouses for flower cultivation, posing a trade-off between agricultural production and environmental concerns, including the asserted need for global land use expansion, and the issues of rural livelihoods and food security [34].

Despite possible drawbacks to the LULC datasets, such as the existence of classification errors and uncertainties [87], its accessibility and availability at different time spans offers considerable advantages for studying land cover changes [88], providing a consistent source of primary data facilitating the reproducibility of results. In addition, post-classification or editing process of vector maps, complemented with the images and analytical capabilities of Google Earth engine allows more accurate identification of distinct land use classes [89].

Regardless of these limitations, we envisage that the proposed DPSIR framework and the practical implementation analysis of LULC transitions and their drivers, using official LULC maps and other freely available databases from distinct sources (demographic, climatic, topographic, etc.), could be replicated to understand environmental change in tropical mountain systems. These types of approaches are particularly important in areas of data scarcity and low technical capacities for the processing of remote sensing information required for land management and planning, which characterizes many distinct territorial levels of governance in tropical mountain systems and developing countries.

The assessment of local and regional patterns of current land use and past land cover conversion is the first step in developing sound land management plans that could prevent broad scale, irreversible ecosystem degradation [77]. This characterization of landscape patterns through time and the analysis of their proximate drivers of landscape change enhance our understanding of how landscape patterns might influence ecosystem services [19]. Our findings would help distinguish important areas for conserving native ecosystems. In addition, our study highlights that research and landscape management, zonation and ecological recovery/restoration should be better integrated into land-use policy and conservation agendas at the local level [77] to balance the multiple needs and benefits from ecosystems of a growing population in the rural landscape of northern Ecuador.

## Conclusions

Our study proposes an adaptation of the DPSIR framework, as a tool to characterize the complexity of tropical mountain systems and conduct integrated ecosystem and ES assessments. After testing the initial phases of the framework in the highlands of northern Ecuador, we present the following conclusions: (1) we found a dynamic and clear geographical pattern of distinct LULC transitions through time. In a span of 24 years, the urban and floriculture zones increased substantially (by 25 and 13 folds, respectively to their original extent, which was less than 2 km$^2$ in 1999); these transitions were observed in the lower elevation bands localized to the east of the study region (less than 2800 m), mainly occupying previous agricultural land. Between 1990 and 2014, the native forests experienced a 40% reduction, with the lowest probability of persistence in the elevation band of 2800–3300 m, where agricultural land and planted forest are continually replacing this LULC class. Our findings also revealed an unexpected stability trend of paramo (0.75–0.90) and a successional recovery of previous agricultural land to the west and center of the territory, which could be explained by agricultural land abandonment. (2) Our conservative results from the GAMs explained between 21 to 42% of the variation of the distinct LULC transitions observed in the study region. Different combination of human induced, and environmental variables were the explanatory driving forces, whereas

topographic factors, resulted in the main drivers of change in this landscape. Interestingly, floricultural expansion was also explained by water availability by irrigation and the production gradient across parishes, whereas shrubland, urban and agricultural transitions can be explained by demographic and infrastructure driving forces, which could be related to urban-rural population dynamics that need further analysis. Future work will include implementing all the phases of the proposed DPSIR framework, which include a multitemporal Ecosystem Service evaluation of the studied landscape.

## Supporting information

**S1 Fig. Transition probability of shrubs and herbs through time in Pedro Moncayo county, by altitudinal bands at the parish level.**
(TIF)

**S2 Fig. Transition probability of páramo through time in Pedro Moncayo county, by altitudinal bands at the parish level.**
(TIF)

**S3 Fig. Spatial distribution of each driver grouping for the first period of analysis.** Each map represents the PC1 from the Principal Component Analysis carried out for each driver of change grouping from period 1 (1990 and 2000).
(TIF)

**S4 Fig. Spatial distribution of each driver grouping for the second period of analysis.** Each map represents the PC1 from the Principal Component Analysis carried out for each driver of change grouping from period 2 (2000 and 1990).
(TIF)

**S5 Fig. Generalized additive model partial dependence plots for forest páramo loss.** Each plot shows a covariate and their partial dependence on probability of páramo loss in the context of the model. The y axis shows the mean of the probability of native forest loss and the x axis the covariate interval. The gray area represents the 95% confidence interval.
(TIF)

**S6 Fig. Generalized additive model partial dependence plots for shrubland loss.** Each plot shows a covariate and their partial dependence on probability of shrubland loss in the context of the model. The y axis shows the mean of the probability of shrubland loss and the x axis the covariate interval. The gray area represents the 95% confidence interval.
(TIF)

**S7 Fig. Generalized additive model partial dependence plots for agricultural transition.** Each plot shows a covariate and their partial dependence on probability of agricultural expansion in the context of the model. The y axis shows the mean of the probability of agricultural expansion and the x axis the covariate interval. The gray area represents the 95% confidence interval.
(TIF)

**S8 Fig. Generalized additive model partial dependence plots for floriculture transition.** Each plot shows a covariate and their partial dependence on probability of floriculture transition in the context of the model. The y axis shows the mean of the probability of floriculture transition and the x axis the covariate interval. The gray area represents the 95% confidence interval.
(TIF)

**S9 Fig. Generalized additive model partial dependence plots for urban transition.** Each plot shows a covariate and their partial dependence on probability of urban transition in the context of the model. The y axis shows the mean of the probability of native forest loss and the x axis the covariate interval. The gray area represents the 95% confidence interval. (TIF)

**S1 Table. Land Use Land Cover (LULC) classification scheme used to assess LULC change analysis [37].** (PDF)

## Acknowledgments

Thanks to the staff of the National Institute of Statistics and Census for their advice in interpreting the coding of the databases. Thanks to Phoebe Lehmann Zarnetske for her sound suggestions on the available climate databases. We are grateful to the authorities and officials of the Autonomous Decentralized Government of Pedro Moncayo for providing georeferenced information of the studied territory. And last but not least, thanks to the students of the Facultad de Ciencias Biológicas de la Universidad Central del Ecuador who helped to collect information, especially Genesis Granja who directly supported the work by obtaining geospatial data related to the drivers of change.

## Author Contributions

**Conceptualization:** Paulina Guarderas, Franz Smith, Marc Dufrene.

**Data curation:** Paulina Guarderas.

**Formal analysis:** Paulina Guarderas, Franz Smith.

**Funding acquisition:** Paulina Guarderas, Marc Dufrene.

**Investigation:** Paulina Guarderas.

**Methodology:** Paulina Guarderas, Franz Smith.

**Project administration:** Paulina Guarderas.

**Resources:** Paulina Guarderas, Franz Smith.

**Software:** Paulina Guarderas, Franz Smith.

**Supervision:** Marc Dufrene.

**Validation:** Paulina Guarderas, Franz Smith, Marc Dufrene.

**Visualization:** Paulina Guarderas, Franz Smith.

**Writing – original draft:** Paulina Guarderas.

**Writing – review & editing:** Paulina Guarderas, Franz Smith, Marc Dufrene.

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
