## [Decision Letter · Decision Letter 0]

20 Dec 2021

PONE-D-21-35047Land use land cover
dynamics through time and their proximate drivers of change in a tropical mountain
system: a case study in a highland landscape of northern
EcuadorPLOS ONE

Dear Dr. Guarderas,

Thank you for submitting your manuscript to PLOS ONE. After careful consideration, we
feel that it has merit but does not fully meet PLOS ONE’s publication criteria as it
currently stands. Therefore, we invite you to submit a revised version of the
manuscript that addresses the points raised during the review process.

I have carefully gone through the comments of reviewers and feel that MS can be
accepted after a major revision.

Please submit your revised manuscript by Feb 03 2022 11:59PM. If you will need more
time than this to complete your revisions, please reply to this message or contact
the journal office at plosone@plos.org. When
you're ready to submit your revision, log on to https://www.editorialmanager.com/pone/ and select the 'Submissions
Needing Revision' folder to locate your manuscript file.

Please include the following items when submitting your revised
manuscript:A rebuttal letter that responds to each point raised by the academic
editor and reviewer(s). You should upload this letter as a separate file
labeled 'Response to Reviewers'.A marked-up copy of your manuscript that highlights changes made to the
original version. You should upload this as a separate file labeled
'Revised Manuscript with Track Changes'.An unmarked version of your revised paper without tracked changes. You
should upload this as a separate file labeled 'Manuscript'.

If you would like to make changes to your financial disclosure, please include your
updated statement in your cover letter. Guidelines for resubmitting your figure
files are available below the reviewer comments at the end of this letter.

We look forward to receiving your revised manuscript.

Kind regards,

Manoj Kumar

Academic Editor

PLOS ONE

Journal Requirements:

3. We note that Figures 1, 2, S3 and S4 in your submission contain [map/satellite]
images which may be copyrighted. All PLOS content is published under the Creative
Commons Attribution License (CC BY 4.0), which means that the manuscript, images,
and Supporting Information files will be freely available online, and any third
party is permitted to access, download, copy, distribute, and use these materials in
any way, even commercially, with proper attribution. For these reasons, we cannot
publish previously copyrighted maps or satellite images created using proprietary
data, such as Google software (Google Maps, Street View, and Earth). For more
information, see our copyright guidelines: http://journals.plos.org/plosone/s/licenses-and-copyright.

a) You may seek permission from the original copyright holder of Figures 1, 2, S3 and
S4 to publish the content specifically under the CC BY 4.0 license.  

Reviewers' comments:

Reviewer's Responses to Questions

**Comments to the Author**

1. Is the manuscript technically sound, and do the data support the conclusions?

Reviewer #1: No

Reviewer #2: No

Reviewer #3: Yes

Reviewer #4: Partly

2. Has the statistical analysis been performed
appropriately and rigorously? 

Reviewer #1: No

Reviewer #2: No

Reviewer #3: Yes

Reviewer #4: Yes

3. Have the authors made all data underlying the
findings in their manuscript fully available?

Reviewer #1: No

Reviewer #2: Yes

Reviewer #3: Yes

Reviewer #4: Yes

4. Is the manuscript presented in an intelligible
fashion and written in standard English?

Reviewer #1: No

Reviewer #2: Yes

Reviewer #3: Yes

Reviewer #4: No

5. Review Comments to the Author

Reviewer #1: A

Analysis, change detection, classification methods, and plant species detection using
Landsat images are not mentioned.

What method is used for primary categorization? What are the indices of validation
and verifications of the categorization? Why is the modified categorization used and
for what parameters (be specific)?

What spectral vegetation index did use to extract plant characteristics and classify
them?

The scale of maps, including MAE maps and the maps taken from Landsat images
analysis, is unknown.

B

The role of stimuli in the spotted changes is not thoroughly covered with regard to
their relationship with each other.

The results of decreasing the dimensions are not thoroughly highlighted and
elaborated for the five mentioned groups.

The authors could use Markov's advanced mode known as Latent MC for use in the real
world.

As a data screening model, the results of the DPSIR model are not expressed for all
variables (drivers) and are not complete.

Why is DPSIR listed in the introduction?

What is the evaluation and policy method that led to the selection of 13 variables
for LULC? Not exactly specified.

C

The article deals with data screening (ecological and socio-economic variables),
evaluation, decision-making, monitoring, and policy-making, but the content is
scattered, inadequate and unreliable. Each of these topics should be defined
separately in its section and the results of each.

Any decision in data mining process shall be based on the type of data and primary
knowledge of the data structure and behavioral pattern. This is not observed in any
of the statistical methods used.

The statistical methods are outdated and the submitted study lacks novelty.

Methodology needs major improvement.

Reviewer #2: Dear Author/s

Congratulations on your good work. The manuscript PONE-D-21-35047 entitled ‘Land use
land cover dynamics through time and their proximate drivers of change in a tropical
mountain system: a case study in a highland landscape of northern Ecuador’ is well
written. On the other hand, there are some essential comments author/s should take
into consideration:

1. Theoretical Framework: No explicit theoretical framework and important theoretical
assumptions is observed?

2. Concepts: A central concept of the study is not adequately and clearly defined? No
new concepts have been added to the discipline.

3. Argument: Central argument is missing? It must be tightly and well written with
examples referring from global, regional and local levels. Lots of studies have been
published in recent years on a similar theme.

4. Literature review and use of references are inadequate. Most of the references are
older than 10 years and only a few latest references are cited. Some recently
updated references need to be added. Following latest references may also be cited
in the manuscript for value addition:

• Mishra, P.K. Rai, A. and Rai, S.C. (2020) Land Use and Land Cover Change Detection
using Geospatial Techniques in Sikkim Himalaya, India. Egyptian Journal of Remote
Sensing and Space Sciences. 23 (2):133-143. doi.org/10.1016/j.ejrs.2019.02.001 IF: 5.18 (2020).

• Mishra, P.K. and Rai, A. (2020) Role of Unmanned Aerial Systems for Natural
Resource Management. Journal of the Indian Society of Remote Sensing. ISSN
0974-3006. DOI: 10.1007/s12524-020-01230-4 (SCOPUS). IF: 1.56 (2020).

5. The title is vague and too lengthy.

6. This study does not provide any new contribution in the field. The study lacks
novelty and continuity in the analysis and provides general findings.

7. Authors have selected “Official Land Use Land Cover (LULC) maps from the Ministry
of Environment of Ecuador (MAE) of four periods of time: 1990, 2000, 2008 and 2014”.
However, the months and percentage of cloud cover available in satellite data are
not mentioned.

8. While conducting LULC analysis, accuracy assessments play a major role in
determining the overall accuracy of results. In the present manuscript accuracy
assessments of maps taken from the Ministry of Environment of Ecuador is not
mentioned.

9. Accuracy assessment of 5 additional topologies produced by authors is also not
available in the manuscript.

10. Authors have mentioned in lines 124-125 that they digitized these 5 topologies
that include planted forests, developed areas (populated zones), horticulture (areas
represented by greenhouses) and natural water bodies. Line 138 authors stated that
“because the study area corresponds to the major centre of floriculture production
for the export market in 138 Ecuador [37,38], we added floriculture crop, as a
separate typology from the agricultural land”. Areas represented by greenhouses can
also have vegetations and other crops in them how was the area under floriculture
estimated?

11. Digitization requires expertise in image interpretation and classes such as
floriculture or horticulture are hard to map on Landsat TM data. And using
Greenhouses as an indicator of horticulture and mapping entire areas of greenhouses
as horticulture or floriculture without surveying is not recommended as many of them
might not be in use. Results of LULC show that there is a huge growth in
floriculture in the study area. However, digitizing greenhouse as floriculture is
not a reliable or accurate method of mapping.

12. No Field data/assessment is available for produced LULC maps.

13. Four periods of time: 1990, 2000, 2008 and 2014 have been selected in the
manuscript. From 2014 to 2021 the world has seen exponential growth in terms of LULC
changes. Why not produce LULC maps of 2020 or 2021 and then do the analysis.

14. If the selected data is almost 7 years old, how do the authors think their
results related to LULC change till 2014 and drivers of change are relevant
today?

15. The whole manuscript is consisting of multiple grammatical and typographical
errors.

16. Data sets chosen for the study has a different spatial resolution for both data.
Landsat 5 (30m) and LISS 3 (23.5m). Rescaling of the data was not mentioned in the
manuscript. Not rescaling both datasets on the same resolution causes inaccuracy in
change detection due to different pixel sizes and even if it’s done the gap of
resolution between two original data sets should be minimum. Here rescaling data
from 30 m to 23.5m or vice versa would cause serious errors/loss of data.

17. Limitations of the study are not mentioned anywhere e.g., areas and reasons for
misclassification.

18. References have to be rechecked as they lack similarity in writing style. In a
few references, the page number is written confusingly.

19. The practical implacability of the study in any other field of work is missing.
Recommendations for Future Work may add value to the manuscript.

20. The authors must check the grammar, consistency and flow of the texts in the
manuscript before submitting for publication.

Reviewer #3: 1. The analysis has been performed between 1990-2014. Unless there is a
very good reason, I would expect to see the analysis to the latest time.

2. L 174-177 Kindly make the explanation and the table 1 consistent

3. There are two tables labelled Table 2 ( L237 and L 320). The text nowhere explains
table 3 even if the later is labelled as table 3.

Reviewer #4: Dear Authors,

Thank you for your manuscript “Land use land cover dynamics through time and their
proximate drivers of change in a tropical mountain system: a case study in a
highland landscape of northern Ecuador”. It covers an interesting topic that fully
matches the scope of the journal “PLOS ONE”. I recommended accepting the topic but
advise a major amendment and improvement of your manuscript. Your manuscript
describes land use land cover dynamics through time and their drivers force of
change. The subject you took is important, the paper nicely written, however I have
some issues to discuss with the authors.

(1) Modify the abstract and add some content showing the methodology applied and the
clear results obtained from the study. The conclusion part of the abstract is
redundant and provides too general a description of results.

(2) The paper has many long sentences, grammar errors and is poorly written.
Extensive English editing is needed.

(3) The Introduction section should be improved, and at the end of the introduction,
you should clearly state the objective. For instance, in your last statement in the
introduction, you have stated that you will employ the DPSIR framework in assessing
the role of different drivers on LULCC in the study area, but it is not featured out
in your methodology. Please clarify this.

(4) In the results section, merge heading “Coverage area for each year” and
“Land-change dynamics through time”

(5) In Table 2. Rename the caption to match the content of the paper

(6) [318]- Renumber the table, is not Table 2 as it is, I think it should refer to
Table 3

(7) [363-366]- not clear what you wanted to say

(8) [392-389]- too long sentence not clear what you wanted to say

(9) [457-462]- English should be checked

(10) [520-526]- too long sentence not clear what you wanted to say

6. PLOS authors have the option to publish the peer
review history of their article (what does this mean?). If published, this will
include your full peer review and any attached files.

If you choose “no”, your identity will remain anonymous but your review may still be
made public.

**Do you want your identity to be public for this peer review?** For
information about this choice, including consent withdrawal, please see our
Privacy Policy.

Reviewer #1: **Yes: **Somayeh Mehrabadi

Reviewer #2: No

Reviewer #3: No

Reviewer #4: No

PONE.docx
---

## [Author Response · Author response to Decision Letter 0]

22 Feb 2022

Responses to the Academic Editor’s suggestions:

Completed. Accordingly with your suggestion, we have carefully checked the PLOS ONE
style templates and we have made adjustments throughout the manuscript to fulfill
the requirements.

2.We note that you have stated that you will provide repository information for your
data at acceptance. Should your manuscript be accepted for publication, we will hold
it until you provide the relevant accession numbers or DOIs necessary to access your
data. If you wish to make changes to your Data Availability statement, please
describe these changes in your cover letter and we will update your Data
Availability statement to reflect the information you provide.

Completed. We have created an account and uploaded our datasets to Zenodo. This is
one of the open access repositories recommended by PLOS ONE. We are providing below
the DOI necessary to access our data:

10.5281/zenodo.5911876

3. We note that Figures 1, 2, S3 and S4 in your submission contain [map/satellite]
images which may be copyrighted. All PLOS content is published under the Creative
Commons Attribution License (CC BY 4.0), which means that the manuscript, images,
and Supporting Information files will be freely available online, and any third
party is permitted to access, download, copy, distribute, and use these materials in
any way, even commercially, with proper attribution. For these reasons, we cannot
publish previously copyrighted maps or satellite images created using proprietary
data, such as Google software (Google Maps, Street View, and Earth). For more
information, see our copyright guidelines: http://journals.plos.org/plosone/s/licenses-and-copyright.

a) You may seek permission from the original copyright holder of Figures 1, 2, S3 and
S4 to publish the content specifically under the CC BY 4.0 license.

Completed. In our revised manuscript we added another Figure (Fig 1) to depict a
Conceptual framework, then, the previous Figures 1 and 2, now correspond to Figures
2 and 3. These two Figures represent maps of: the study area (new Fig 2) and the
Land Use Land Cover changes through time (new Fig 3). As you suggested, we properly
referred the data sources of each map in the Figure’s captions and within the map’s
labels.

In addition, we searched in the journal’s repository for other articles that used
georeferenced data from the official sources from Ecuador to portray maps, and we
found this recent publication: Ortega-Andrade HM, Rodes Blanco M, Cisneros-Heredia
DF, Guerra Arévalo N, López de Vargas-Machuca KG, Sánchez-Nivicela JC, et al. (2021)
Red List assessment of amphibian species of Ecuador: A multidimensional approach for
their conservation. PLoS ONE 16(5): e0251027. https://doi.org/10.1371/journal.pone.0251027. Then, we used this
article as an example to follow, regarding the way the similar data sources were
referred and cited.

We have also sent official communications to the Ecuadorian Governmental Institutions
where geospatial data was obtained from (Ministerio del Ambiente, Agua y Transición
Ecológica, y al Instituto Geográfico Militar) to ask information regarding the
freely available data that we downloaded and their licenses, but we have not had had
a response yet. Therefore, we cite the data sources directly, as Ortega-Andrade HM,
Rodes Blanco M, Cisneros-Heredia DF, Guerra Arévalo N, López de Vargas-Machuca KG,
Sánchez-Nivicela JC, et al. (2021). 

We included both in the Methods and within the map’s labels of each Figure the public
available URL of the web sites of the Ecuadorian Governmental Institutions where
data was download for depicting the maps and running our analysis.

Two photos included in the new Figure 6 were obtained from Google Maps, and now they
were replaced. Additionally, regarding the photos included in the new Figure 4, all
were taken by the main author of this manuscript. So, we consider a written
permission is not required (but if it is needed, she is willing to sign and submit
the document).

Finally, Figures S3 and S4 portray results of the Principal Component Analysis of the
drivers of change conducted in this research. These results were merged to the
shapefiles of the census areas and spatially represented the results of our study.
Therefore, they are not copyrighted figures. And the sources of information are
properly cited in the Figure’s caption and elsewhere in the text.

Responses to the reviewer’s general observation and questions

1. Is the manuscript technically sound, and do the data support the conclusions?

Reviewer #1: No

Reviewer #2: No

Reviewer #3: Yes

Reviewer #4: Partly

We have improved the theoretical and the methodological framework of the manuscript
to better connect all the elements of our study. Our previous draft did not include
the Conclusion section explicitly, therefore, our revised version includes a
detailed section with the main conclusions drawn from our analysis.

2. Has the statistical analysis been performed appropriately and rigorously?

Reviewer #1: No

Reviewer #2: No

Reviewer #3: Yes

Reviewer #4: Yes

As mentioned in our previous response, we consider that with a better description of
the conceptual and methodological framework presented in the revised manuscript, the
further statistical analyses are well supported and are presented in a more
consistent and clearer manner. Following this scheme, the results and conclusions
easily follow our framework. 

3. Have the authors made all data underlying the findings in their manuscript fully
available?

The [ http://www.plosone.org/static/policies.action#sharing | PLOS Data
policy ] requires authors to make all data underlying the findings described in
their manuscript fully available without restriction, with rare exception (please
refer to the Data Availability Statement in the manuscript PDF file). The data
should be provided as part of the manuscript or its supporting information, or
deposited to a public repository. For example, in addition to summary statistics,
the data points behind means, medians and variance measures should be available. If
there are restrictions on publicly sharing data—e.g. participant privacy or use of
data from a third party—those must be specified.

Reviewer #1: No

Reviewer #2: Yes

Reviewer #3: Yes

Reviewer #4: Yes

As mentioned in our response to the second comment from the General Editor, we have
made all our data fully available in the Zenodo’s open access repository. This is
the DOI 10.5281/zenodo.5911876 to access our datasets.

4. Is the manuscript presented in an intelligible fashion and written in standard
English?

Reviewer #1: No

Reviewer #2: Yes

Reviewer #3: Yes

Reviewer #4: No

The manuscript was revised by English native speakers from an editing and
proofreading service. Then, the English grammar of the updated version has been
greatly improved.

Responses to Specific comments from different reviewers:

Specific comments from Reviewer #1:

A. Analysis, change detection, classification methods, and plant species detection
using Landsat images are not mentioned.

What method is used for primary categorization? 

Completed. We have added in the methods a more detailed explanation of the
classification process used by the team of geographers from the Ministry of
Environment and the Ministry of Agriculture, Livestock, Aquaculture and Fisheries of
Ecuador to produce the LULC maps used in this article. Briefly, to generate the LULC
coverages they used a supervised classification method with training data of regions
of interest (ROis), using the maximum likelihood clustering algorithm. The
supervised classification was performed in the ENVI software with automated image
classification tools that allow the use of previously defined and refined areas or
regions ROIs. 

More details on the processing and classification methods used by MAE and MAGAP can
be found here:

MAE, MAGAP. Protocolo metodológico para la elaboración del mapa de cobertura y uso de
la tierra del Ecuador Continental. 2015. 

MAE. Análisis de la deforestación en el Ecuador Continental 1990 - 2014. [Internet].
Quito - Ecuador; 2016. Available from: http://suiadoc.ambiente.gob.ec/documents/10179/1149768/AnalisisDeforestacionEcuador1990_2014.pdf/8285da57-c6ca-4e82-9be7-ccea3c9317cb

What are the indices of validation and verifications of the categorization? 

To answer the reviewer’s inquiry, an expanded explanation was added in the Methods
section. 

Briefly, the accuracy assessment analysis of the official LULC maps encompassed an
independent interpretation process, where an experienced image expert classified the
coverage for each sample obtained from a stratified random sampling protocol,
according to the methods proposed in the FRA global remote sensing survey (Forestry
Department, 2009); this was done based on the Level 1 legend of the LULC maps of
Continental Ecuador. A confusion matrix was created using the JRC Validation Tool
program for the sampling areas (Simonetti, Beuchle, & Eva, 2011), and the
following overall accuracy values were obtained: 69%, 73%, 76% and 85% for the years
1990, 2000, 2008 and 2014, respectively (MAE, 2016).

More details on the verification methods used by MAE and MAGAP can be found here: 

Forestry Department. (2009). The 2010 global forest resources assessment remote
sensing survey: an outline of the objectives, data, methods and approach. Forest
Resources Assessment Working Paper.

Simonetti, D., Beuchle, R., & Eva, H. (2011). User Manual for the JRC Land
Cover/Use Change Validation Tool. Retrieved from https://publications.jrc.ec.europa.eu/repository/handle/JRC62603

MAE. (2016). Análisis de la deforestación en el Ecuador Continental 1990 - 2014.
Retrieved from http://suiadoc.ambiente.gob.ec/documents/10179/1149768/AnalisisDeforestacionEcuador1990_2014.pdf/8285da57-c6ca-4e82-9be7-ccea3c9317cb

Why is the modified categorization used and for what parameters (be specific)?

We clarified and expanded the Method’s section, explaining the reviewer’s
question.

In essence, to refine the results and obtain a more accurate map of the study area, a
thorough editing process of the official vector maps from the different study
periods was carried out. To support this editing process, as carried out by
(Madrigal-Martínez & Miralles i García, 2019), distinct secondary sources of
information were revised such as field points, Google Earth images, orthophotographs
and other official such as the ecosystem coverage (http://ide.ambiente.gob.ec/mapainteractivo/), the floriculture
cadastral and other maps from the Ministry of Agriculture (http://geoportal.agricultura.gob.ec/). In
addition, composite LANDSAT images from our study area, using radiometric
enhancements and spectral band combinations were also used (Gorelick et al., 2017).
From the editing process, mainly five typologies were improved. These included:
planted forests, developed areas (populated zones), floriculture (areas represented
by greenhouses) and natural water bodies. Following the methods proposed by (Jin,
Ismail, Muharam, & Alias, 2021), a point-based accuracy assessment was conducted
using Google Earth as a verification source. 

Madrigal-Martínez, S., & Miralles García, J. L. (2019). Land-change dynamics and
ecosystem service trends across the central high-Andean Puna. Scientific Reports,
9(1), 1–12. https://doi.org/10.1038/s41598-019-46205-9

Gergel, S. E., & Turner, M. G. (Eds.). (2017). Learning landscape ecology: a
practical guide to concepts and techniques. Springer.

Jin, D. H., Ismail, M. H., Muharam, F. M., & Alias, M. A. (2021). Evaluating the
impacts of land use/land cover changes across topography against land surface
temperature in Cameron Highlands. PLoS ONE, 16(5 May), 1–26. https://doi.org/10.1371/journal.pone.0252111

What spectral vegetation index did use to extract plant characteristics and classify
them?

Spectral vegetation indexes were not used by the project team from MAE and MAGAP for
the Land Use Land Cover classification, instead, a supervised process with RBD
signatures for LULC classes in the ENVI software. See our two first responses in
section A for a further explanation.

The scale of maps, including MAE maps and the maps taken from Landsat images
analysis, is unknown.

Completed, This is detailed in the revised methods. Resulting maps were produced in a
mapping scale of 1:100,000 with Landsat (TM) images as main inputs.

B. The role of stimuli in the spotted changes is not thoroughly covered with regard
to their relationship with each other.

The results of decreasing the dimensions are not thoroughly highlighted and
elaborated for the five mentioned groups.

Completed. Following the reviewer’s comment, we have added in the Methods an
explanation of the results from the dimension reduction after conducting the PCAs
within each grouping driver.

The authors could use Markov's advanced mode known as Latent MC for use in the real
world.

We did not use Latent Markov Models, which is a very interesting conceptual and
methodological approach to explore ecological systems. However, we used the
transition probabilities obtained in Markov chain analyses and integrate them into
another very powerful statistical model (General Additive Model), currently in use
by ecologists (Woods 2017a), to reveal important drivers of change for land use land
cover dynamics. In that sense, we consider it as an alternative to LMM and an
innovative inferential method that seek to uncover the relationships between factors
driving dynamics in ecological systems and thereby predict them in quantitative
terms.

As a data screening model, the results of the DPSIR model are not expressed for all
variables (drivers) and are not complete.

In response to this comment, we included a graphical scheme (new Fig 1) and a
detailed description of our proposed conceptual framework in the Method’s section.
Afterwards, we clearly described the way we implemented the DPSIR framework,
explaining how we are conceiving the driving forces and their effect on LULC
transitions. Subsequently, we described the analytical approach to include different
drivers of change within groups, the way we conducted variable screening and the
dimension reduction process, to finally present the statistical model that we used
to test the driving forces that explain LULC changes in our case study.

Why is DPSIR listed in the introduction?

Since we are proposing an adapted DPSIR approach for tropical mountain systems and we
implemented an initial assessment using this approach in a case study in the
highlands of Ecuador, we consider important to expand the description of the DPSIR
approach in the Introduction. 

For a more detailed explanation of the proposed conceptual framework, please see our
response to the second reviewer's first comment. 

What is the evaluation and policy method that led to the selection of 13 variables
for LULC? Not exactly specified.

In the Methods we added a detailed explanation of the criteria used for selecting the
variables included in the model to explain LULC change, and the way we fulfilled
each selection criteria. 

Briefly, we followed the criteria selection in the context of a DPSIR framework, as
proposed by: Wang Z, Zhou J, Loaiciga H, Guo H, Hong S. A DPSIR model for ecological
security assessment through indicator screening: A case study at Dianchi Lake in
China. PLoS One. 2015;10(6):1–13. These authors, propose the following criteria:
‘(1) Relevancy: indicators should reflect the underlying cause of environmental
change. (2) Availability: the indicator data should be available, accessible, and
consistent within the period of analysis. (3) Independence: indicators must be
independent of each other to eliminate multicollinearity. (4) Representativeness:
each indicator used in the model must represent a category or phenomenon of its own,
and must provide superior information to other indicators in a similar category’
(p.4).

C. The article deals with data screening (ecological and socio-economic variables),
evaluation, decision-making, monitoring, and policy-making, but the content is
scattered, inadequate and unreliable. Each of these topics should be defined
separately in its section and the results of each.

Any decision in data mining process shall be based on the type of data and primary
knowledge of the data structure and behavioral pattern. This is not observed in any
of the statistical methods used.

As mentioned in our previous response, our updated manuscript provides an improved
explanation of the connections between the Drivers of change and the actual
pressures on the environment in the context of the adapted DPSIR framework. We
consider that our current explanation of the framework set the stage to better
understand the methods and the results presented in our article. Here, we proposed
an adaptation of the whole DPSIR framework in the context of tropical mountain
systems, with all the 5 elements of the conceptual model.

We also clarified that the purpose of the current paper is to develop an initial
ecosystem assessment including only Drivers and Pressures. A further ecosystem
assessment, to complete the assessment with all 5 elements of the DPSIR, is under
work in our case study.

Additionally, we added a section to describe the results of our data screening and
the dimension reduction after conducting Principal Component Analysis (PCAs), in
each group of the tested drivers of change.

The statistical methods are outdated and the submitted study lacks novelty. 

We consider that our study is unique in different aspects. Firstly, it adapts the
DPSIR holistic approach to the context of tropical mountain systems and implement
the first elements of the framework in a sensitive region of the northeastern
Ecuadorian Andes. Then, it contributes to identify the key characteristics of
tropical mountain social-ecological systems that should be represented in ecosystem
assessments to support evidence-based policy and management actions. We connected
our proposed conceptual framework with a step by step methodological approach, which
was initially implemented in a case study located in the highlands of northern
Ecuador

Secondly, we portrayed land use land cover (LULC) dynamics using Markov-chain
probabilities by elevation and geographic settings, which is an interesting
characterization, recently conducted in other regions but limited for the highland
landscapes in the Andes. In addition, we consider that Markov chain analysis is a
well-known approach to describe LULC transitions, currently in use as supported by
the following recent publications:

Gergel, S. E., & Turner, M. G. (Eds.). (2017). Learning landscape ecology: a
practical guide to concepts and techniques. Springer.

Hamad, R., Balzter, H., & Kolo, K. (2018). Predicting land use/land cover changes
using a CA-Markov model under two different scenarios. Sustainability (Switzerland),
10(10), 1–23. https://doi.org/10.3390/su10103421

Kumar, S., Radhakrishnan, N., & Mathew, S. (2014). Land use change modelling
using a Markov model and remote sensing. Geomatics, Natural Hazards and Risk, 5(2),
145–156. https://doi.org/10.1080/19475705.2013.795502

Liping, C., Yujun, S., & Saeed, S. (2018). Monitoring and predicting land use and
land cover changes using remote sensing and GIS techniques—A case study of a hilly
area, Jiangle, China. PLoS ONE, 13(7), 1–23. https://doi.org/10.1371/journal.pone.0200493

Finally, we integrated Markov’s transitions probabilities (as a response variable)
with a variety of freely available geospatial and temporal data into Generalized
Additive Models (GAMs) to uncover the factors driving such landscape dynamics in a
sensitive region of the northern Ecuadorian Andes. We used GAM regressions to
elucidate two types of transitions estimated through Markov chain analysis: 1) the
probability of natural ecosystems loss, and 2) the probability to change towards
anthropic environments.

GAMs are very powerful statistical models, used extensively in environmental
modelling and provide great scope to model complex relationships between covariates,
as is exemplified in the following references: 

Liping, C., Yujun, S., & Saeed, S. (2018). Monitoring and predicting land use and
land cover changes using remote sensing and GIS techniques—A case study of a hilly
area, Jiangle, China. PLoS ONE, 13(7), 1–23. https://doi.org/10.1371/journal.pone.0200493

Wood, S. (2017). Generalized Additive Models: An Introduction with R. (2nd ed.). CRC
Press.

Methodology needs major improvement.

Based on different suggestions and inquiries from all the reviewers, we have improved
our text in the Methodology to better explain all the steps that we conducted to
fulfill our objectives. We consider that this section from the updated version of
the manuscript has largely improved. 

Specific comments from Reviewer #2:

1. Theoretical Framework: No explicit theoretical framework and important theoretical
assumptions is observed?

As mentioned in our response to comment B from the first reviewer, we are proposing
an adapted Driver Pressure State Impact Response (DPSIR) framework for tropical
mountain systems, and it was not fully developed before. Therefore, for our updated
manuscript we have highlighted this contribution in the Introduction section.
Further, we described that for the purpose of the current article we are developing
an initial ecosystem assessment in the context of the proposed DPSIR, to uncover the
driving forces that exert pressure on the landscapes, by examining LULC changes. In
the Methods, we included a graphical scheme (new Fig 1) and a detailed description
of our proposed conceptual framework. Afterwards, we clearly described the way we
implemented the DPSIR framework, explaining how we are conceiving the driving forces
and their effect on LULC transitions. Subsequently, we described the analytical
approach to include different drivers of change within groups, the way we conducting
variables screening and the dimension reduction process, and present the statistical
model that we used to test the driving forces that explain LULC changes in our case
study.

2. Concepts: A central concept of the study is not adequately and clearly defined? No
new concepts have been added to the discipline.

Completed. We have rewritten much of the introduction to include and highlight key
concepts to describe the issues and focus of the article (eg. land use, land cover,
ecosystem services).

3. Argument: Central argument is missing? It must be tightly and well written with
examples referring from global, regional and local levels. Lots of studies have been
published in recent years on a similar theme. 

We have rewritten the Introduction to better present the central argument, supported
by more recent publications

4. Literature review and use of references are inadequate. Most of the references are
older than 10 years and only a few latest references are cited. Some recently
updated references need to be added. Following latest references may also be cited
in the manuscript for value addition:

• Mishra, P.K. Rai, A. and Rai, S.C. (2020) Land Use and Land Cover Change Detection
using Geospatial Techniques in Sikkim Himalaya, India. Egyptian Journal of Remote
Sensing and Space Sciences. 23 (2):133-143. doi.org/10.1016/j.ejrs.2019.02.001 IF: 5.18 (2020).

• Mishra, P.K. and Rai, A. (2020) Role of Unmanned Aerial Systems for Natural
Resource Management. Journal of the Indian Society of Remote Sensing. ISSN
0974-3006. DOI: 10.1007/s12524-020-01230-4 (SCOPUS). IF: 1.56 (2020).

We have carefully reviewed and updated our references. 

5. The title is vague and too lengthy.

We have revised the title to better summarise the focus of the article

6. This study does not provide any new contribution in the field. The study lacks
novelty and continuity in the analysis and provides general findings. 

We consider that our study is unique in different aspects. We presented a detailed
response to similar issues in our response to comment C from the first reviewer.
Please refer to this previous section.

7. Authors have selected “Official Land Use Land Cover (LULC) maps from the Ministry
of Environment of Ecuador (MAE) of four periods of time: 1990, 2000, 2008 and 2014”.
However, the months and percentage of cloud cover available in satellite data are
not mentioned.

Completed. We have added a detailed description of the classification methods
conducted by the project team from MAE and MAGAP to produce the official LULC maps.
We explained that to optimize the use of cloud-free satellite images to generate the
coverage map for each year, a range of 12 months before and after the defined year
was considered

8. While conducting LULC analysis, accuracy assessments play a major role in
determining the overall accuracy of results. In the present manuscript accuracy
assessments of maps taken from the Ministry of Environment of Ecuador is not
mentioned.

Completed. We have added an explanation in the Methods section. The accuracy
assessment analysis implemented an independent interpretation process, where an
experienced image expert classified the coverage for each sample obtained from a
stratified random sampling protocol, according to the methods proposed in the FRA
global remote sensing survey (Forestry Department, 2009); this was done based on the
Level 1 legend of the LULC maps of Continental Ecuador. A confusion matrix was
created using the JRC Validation Tool program for the sampling areas (Simonetti et
al., 2011), and the following overall accuracy values were obtained: 69%, 73%, 76%
and 85% for the years 1990, 2000, 2008 and 2014, respectively (MAE, 2016).

9. Accuracy assessment of 5 additional topologies produced by authors is also not
available in the manuscript.

We added and explanation of our accuracy assessment in the Methodology as
follows:

Following the methods proposed by (Jin et al., 2021), a point-based accuracy
assessment was conducted using Google Earth as a verification source. After that, a
confusion matrix was created using 600 random points obtained from a stratify
sampling scheme over the altitudinal bands. The resulting overall accuracy of the
edited maps ranged from 82 to 86%. The editing process, using visual digitalization,
over the LULC official vector layers from the periods of interest and the accuracy
assessment were conducted in QGIS 3.10 (QGIS Development Team., n.d.) 

Jin, D. H., Ismail, M. H., Muharam, F. M., & Alias, M. A. (2021). Evaluating the
impacts of land use/land cover changes across topography against land surface
temperature in Cameron Highlands. PLoS ONE, 16(5 May), 1–26.

10. Authors have mentioned in lines 124-125 that they digitized these 5 topologies
that include planted forests, developed areas (populated zones), horticulture (areas
represented by greenhouses) and natural water bodies. Line 138 authors stated that
“because the study area corresponds to the major centre of floriculture production
for the export market in 138 Ecuador [37,38], we added floriculture crop, as a
separate typology from the agricultural land”. Areas represented by greenhouses can
also have vegetations and other crops in them how was the area under floriculture
estimated?

We have clarified and corrected the main text. The previous manuscript was incorrect
in describing that the floriculture crop as a separate typology from the
agricultural land typology; instead in the LULC official maps, it was included as
part of the infrastructure, within the developed areas. We corrected the text and
added some more detail on the characteristics of floriculture production.

We also expanded the justification to assume that greenhouses in our study region
were considered for floriculture production

11. Digitization requires expertise in image interpretation and classes such as
floriculture or horticulture are hard to map on Landsat TM data. And using
Greenhouses as an indicator of horticulture and mapping entire areas of greenhouses
as horticulture or floriculture without surveying is not recommended as many of them
might not be in use. Results of LULC show that there is a huge growth in
floriculture in the study area. However, digitizing greenhouse as floriculture is
not a reliable or accurate method of mapping.

We have clarified and expanded the description of our editing process, where we used
different sources to improve the classification such as the floriculture cadastral
map from the geoportal of the Ministry of Agriculture.

Additionally, it is important to point out that a field study, complementary to this
landscape analysis is being conducted by the main author, to understand the effect
of land use on biodiversity and ecosystem services, therefore, regular field visits
have been carried out, providing the author a good geographic understanding of the
current situation of the study area

12. No Field data/assessment is available for produced LULC maps.

We have described in the Methods the assessment conducted by MAE and MAGAP to produce
the official LULC maps. A similar inquiry was posed by the first reviewer. Please
see our detailed response for the second comment of reviewer #1.

In addition, a point-based accuracy assessment was conducted using the historical
images from Google Earth as a verification source for the final maps obtained after
our editing process in our study region. This is an alternative method to field data
verification and assessment as proposed by: Jin, D. H., Ismail, M. H., Muharam, F.
M., & Alias, M. A. (2021). Evaluating the impacts of land use/land cover changes
across topography against land surface temperature in Cameron Highlands. PLoS ONE,
16(5 May), 1–26.

13. Four periods of time: 1990, 2000, 2008 and 2014 have been selected in the
manuscript. From 2014 to 2021 the world has seen exponential growth in terms of LULC
changes. Why not produce LULC maps of 2020 or 2021 and then do the analysis.

We decided to use official LULC maps and other freely available databases from
distinct sources (demographic, climatic, topographic, etc.) as a practical DPSIR
implementation example that could be replicated to understand environmental change
(LULC transitions and their drivers) in a context of data scarcity and low technical
capacities for the processing of remote sensing information required for land
management and planning. This corresponds to the reality at distinct territorial
level of governance in developing countries.

For our LULC analysis we only included the four periods of time 1990, 2000, 2008 and
2014 from which available and comparable maps were available when the analysis took
place

14. If the selected data is almost 7 years old, how do the authors think their
results related to LULC change till 2014 and drivers of change are relevant
today?

We consider the LULC trends observed in our landscape territory up to 2014 provides a
recent characterization of the trajectories that are occurring in northern Ecuador.
We included information until this year of analysis because it was available when
the study was conducted

15. The whole manuscript is consisting of multiple grammatical and typographical
errors.

Completed. As mentioned before. We already overcome this problem. The manuscript was
revised by English native speakers from an editing and proofreading service. 

16. Data sets chosen for the study has a different spatial resolution for both data.
Landsat 5 (30m) and LISS 3 (23.5m). Rescaling of the data was not mentioned in the
manuscript. Not rescaling both datasets on the same resolution causes inaccuracy in
change detection due to different pixel sizes and even if it’s done the gap of
resolution between two original data sets should be minimum. Here rescaling data
from 30 m to 23.5m or vice versa would cause serious errors/loss of data.

The classification analysis performed by the team of geographers from MAE and MAGAP
used primarily LANDSAT images from different years at a spatial resolution of
30m.

LISS satellite images were not used

17. Limitations of the study are not mentioned anywhere e.g., areas and reasons for
misclassification.

Following the suggestion from the reviewer, we have expanded the limitation section
in our discussion. We expanded the methodological limitations that could led to
misclassification of the LULC maps

18. References have to be rechecked as they lack similarity in writing style. In a
few references, the page number is written confusingly.

Completed. We have carefully checked the references throughout the text and in the
list of references to fulfill the PLOS ONE style. 

19. The practical implacability of the study in any other field of work is missing.
Recommendations for Future Work may add value to the manuscript.

Expand the recommendations focusing in the implacability of our findings.

20. The authors must check the grammar, consistency and flow of the texts in the
manuscript before submitting for publication.

Completed. The manuscript was revised by English native speakers from an editing and
proofreading service on language improvement

Specific comments from Reviewer #3:

1. The analysis has been performed between 1990-2014. Unless there is a very good
reason, I would expect to see the analysis to the latest time.

As it was explained in the comment N. 13 from Reviewer N.2, we decided to use
official LULC maps and other freely available databases from distinct sources
(demographic, climatic, topographic, etc.) as a practical DPSIR implementation
example that could be replicated to understand environmental change (LULC
transitions and their drivers) in tropical mountain systems. These regions, mostly
situated in developing countries, are characterized by a context of data scarcity
and low technical capacities for the processing of remote sensing information
required for land management and planning.

Specifically, this situation depicts the situation from our study region.

For our LULC analysis we only included the four periods of time 1990, 2000, 2008 and
2014 from which available and comparable maps were available when the analysis took
place.

2. L 174-177 Kindly make the explanation and the table 1 consistent

Completed. We have made all the terminology of the driving forces consistent between
the text and Table 1. Additionally, we added a description of the variables included
within each grouping driver.

3. There are two tables labelled Table 2 ( L237 and L 320). The text nowhere explains
table 3 even if the later is labelled as table 3.

Completed. We have corrected the error. 

Specific comments from Reviewer #4:

1) Modify the abstract and add some content showing the methodology applied and the
clear results obtained from the study. The conclusion part of the abstract is
redundant and provides too general a description of results.

Completed. Accordingly with the reviewer’s suggestion, the contextual and
methodological framework was added to the abstract. We have also synthesized the
conclusion and provided more specific results.

2) The paper has many long sentences, grammar errors and is poorly written. Extensive
English editing is needed.

Completed. We already overcome this problem. The manuscript was revised by English
native speakers from an editing and proofreading service. 

3) The Introduction section should be improved, and at the end of the introduction,
you should clearly state the objective. For instance, in your last statement in the
introduction, you have stated that you will employ the DPSIR framework in assessing
the role of different drivers on LULCC in the study area, but it is not featured out
in your methodology. Please clarify this.

Completed.. In the revised introduction we have better explained the DPSIR approach,
as our guiding conceptual framework, which connects Drivers and Pressures in a
landscape pattern analysis, additionally the DPSIR framework was properly
operationalized in the methods. 

We further elaborated on this subject on comment

(4) In the results section, merge heading “Coverage area for each year” and
“Land-change dynamics through time”

Completed. We included the reviewer´s suggestion

(5) In Table 2. Rename the caption to match the content of the paper

Completed. We have corrected the error 

(6) [318]- Renumber the table, is not Table 2 as it is, I think it should refer to
Table 3

Completed. We have corrected the error 

(7) [363-366]- not clear what you wanted to say

Completed. We have rewritten the paragraph to clarify the main point.

8) [392-389]- too long sentence not clear what you wanted to say

We have edited this paragraph following the reviewer’s suggestion.

(9) [457-462]- English should be checked

As mentioned elsewhere. We have corrected this problem with the aid of an Editing
service for proofreading our revised draft.

10) [520-526]- too long sentence not clear what you wanted to say

We have edited this paragraph following the reviewer’s suggestion.

to Reviewers.docx
---

## [Decision Letter · Decision Letter 1]

7 Apr 2022

PONE-D-21-35047R1Land use and land cover
change in a tropical mountain landscape of northern Ecuador: altitudinal patterns
and driving forcesPLOS ONE

Dear Dr. Guarderas,

Thank you for submitting your manuscript to PLOS ONE. After careful consideration, we
feel that it has merit but does not fully meet PLOS ONE’s publication criteria as it
currently stands. Therefore, we invite you to submit a revised version of the
manuscript that addresses the points raised during the review process.

Please submit your revised manuscript by May 22 2022 11:59PM. If you will need more
time than this to complete your revisions, please reply to this message or contact
the journal office at plosone@plos.org. When
you're ready to submit your revision, log on to https://www.editorialmanager.com/pone/ and select the 'Submissions
Needing Revision' folder to locate your manuscript file.

Please include the following items when submitting your revised
manuscript:A rebuttal letter that responds to each point raised by the academic
editor and reviewer(s). You should upload this letter as a separate file
labeled 'Response to Reviewers'.A marked-up copy of your manuscript that highlights changes made to the
original version. You should upload this as a separate file labeled
'Revised Manuscript with Track Changes'.An unmarked version of your revised paper without tracked changes. You
should upload this as a separate file labeled 'Manuscript'.If you would like to make changes to your financial disclosure,
please include your updated statement in your cover letter. Guidelines for
resubmitting your figure files are available below the reviewer comments at the end
of this letter.

We look forward to receiving your revised manuscript.

Kind regards,

Manoj Kumar

Academic Editor

PLOS ONE

Journal Requirements:

Reviewers' comments:

Reviewer's Responses to Questions

**Comments to the Author**

1. If the authors have adequately addressed your comments raised in a previous round
of review and you feel that this manuscript is now acceptable for publication, you
may indicate that here to bypass the “Comments to the Author” section, enter your
conflict of interest statement in the “Confidential to Editor” section, and submit
your "Accept" recommendation.

Reviewer #1: All comments have been addressed

Reviewer #2: All comments have been addressed

Reviewer #3: All comments have been addressed

Reviewer #4: All comments have been addressed

2. Is the manuscript technically sound, and do the data
support the conclusions?

Reviewer #1: Partly

Reviewer #2: Yes

Reviewer #3: Yes

Reviewer #4: Partly

3. Has the statistical analysis been performed
appropriately and rigorously? 

Reviewer #1: N/A

Reviewer #2: Yes

Reviewer #3: Yes

Reviewer #4: Yes

4. Have the authors made all data underlying the
findings in their manuscript fully available?

Reviewer #1: Yes

Reviewer #2: Yes

Reviewer #3: Yes

Reviewer #4: Yes

5. Is the manuscript presented in an intelligible
fashion and written in standard English?

Reviewer #1: Yes

Reviewer #2: Yes

Reviewer #3: Yes

Reviewer #4: Yes

6. Review Comments to the Author

Reviewer #1: After reviewing the revised file submitted by the authors, compliance
with the requirements in the initial review phase is generally approved. However,
the following details have been received less attention.

Remote sensing experts believe that accuracy of over 85% is acceptable for detecting
changes in satellite imagery. Hene, reputable papers from remote sensing experts
should be studied in this regard. Assuming that the number and dispersion of
sampling points (Rois) were observed, the accuracy of the article is not
satisfactory. However, with a little leeway, the accuracy of 69-73 and 79 can be
accepted.

To study the plant and agricultural species, a scale of 1.100000 is small. This scale
is more suitable for maps obtained from low-resolution satellite images and for
studying the regional climate.

Google Earth is not a good source for specific remote sensing tasks. It is more
reliable to use mapping organization maps, aerial mapping, or field fraternity
mapping operations.

Reviewer #2: I have reviewed the manuscript number PONE-D-21-35047 full title ‘Land
use and land cover change in a tropical mountain landscape of northern Ecuador:
altitudinal patterns and driving forces’. The revised manuscript is improved in
quality and readability. On the other hand, I still have few suggestions author/s
should take into consideration:

1. Title should be ‘Land use and land cover change in a tropical mountain landscape
of northern Ecuador’.

2. Abstract: Generalized Additive Model (GAM) (Line 32) and Multiregression models
(GAM) (Line 41) is creating confusion. Please use different abbreviation for two
models.

3. Suggested references are missing in the revised manuscript reference section
kindly add for value adition.

• Mishra, P.K. Rai, A. and Rai, S.C. (2020) Land Use and Land Cover Change Detection
using Geospatial Techniques in Sikkim Himalaya, India. Egyptian Journal of Remote
Sensing and Space Sciences. 23 (2):133-143. doi.org/10.1016/j.ejrs.2019.02.001

• Mishra, P.K. and Rai, A. (2020) Role of Unmanned Aerial Systems for Natural
Resource Management. Journal of the Indian Society of Remote Sensing. ISSN
0974-3006. DOI: 10.1007/s12524-020-01230-4

4. The practical implacability of the study in any other field of work is still
missing. Recommendations for Future Work may add value to the manuscript.

Best Wishes!

Reviewer #3: The authors have provide enough explanation to my concern. I still
wished they had the latest year covered in the analysis but I do not seek that as
the sole reason to reject it. Hence, I recommend for acceptance to publish provided
that it meets other publication criteria as per the guideline ( copy right and
others).

Reviewer #4: Dear Authors,

Thank you for your manuscript “Land use and land cover change in a tropical mountain
landscape of northern Ecuador: altitudinal patterns and driving forces”. The title
is now well written covering the content of the paper. I recommended accepting the
topic as it fully matches the scope of the journal “PLOS ONE”. Also, now authors
have incorporated most reviewers’ comments provided to them making the manuscript
much better. But there are some issues that need to be addressed by the authors, see
the attached track change word document.

7. PLOS authors have the option to publish the peer
review history of their article (what does this mean?). If published, this will
include your full peer review and any attached files.

If you choose “no”, your identity will remain anonymous but your review may still be
made public.

**Do you want your identity to be public for this peer review?** For
information about this choice, including consent withdrawal, please see our
Privacy Policy.

Reviewer #1: No

Reviewer #2: **Yes: **Dr. Prabuddh Kumar Mishra

Reviewer #3: No

Reviewer #4: **Yes: **Nangware Kajia Msofe

PONE.docx

---

## [Author Response · Author response to Decision Letter 1]

28 Apr 2022

April 28, 2022 

Dear Dr. Manoj Kumar

Academic Editor

PLOS ONE

Thank you for the opportunity to resubmit a revised version of the manuscript
PONE-D-21-35047R1, titled “Land use and land cover change in a tropical mountain
landscape of northern Ecuador: altitudinal patterns and driving forces“ to PLOS
ONE

My co-authors and I have read in detail your email and all the comments from the
reviewers, and when specific suggestions were included we have been able to
incorporate changes in the manuscript, which have been highlighted in the
manuscript. In contrast, for other general comments we have explained our responses
in the rebuttal letter.

Here is a point-by-point response to your suggestions and to the reviewers’ comments
and concerns.

Responses to the Academic Editor’s suggestions:

1. Journal Requirements:

Completed. Accordingly with your suggestion, we have carefully reviewed the reference
list and made some adjustments described as follow:

We correctly placed the references n. 20 and 21 as follow:

Reference n. 20: Brandt JS, Townsend PA. Land use - Land cover conversion,
regeneration and degradation in the high elevation Bolivian Andes. Landsc Ecol.
2006;21(4):607–23)

was moved to line 84.

The new reference n. 21: Vanacker V, Molina A, Torres R, Calderon E, Cadilhac L.
Challenges for research on global change in mainland Ecuador. Neotrop Biodivers
[Internet]. 2018;4(1):114–8. Available from: doi: 10.1080/23766808.2018.1491706 was
added to line 87 was moved to line 87.

Based on one suggestion from the Reviewer#2, we added the reference n.78 in the
Discussion (lines 578, 714 and 719):

Mishra, P.K. Rai, A. and Rai, S.C. (2020) Land Use and Land Cover Change Detection
using Geospatial Techniques in Sikkim Himalaya, India. Egyptian Journal of Remote
Sensing and Space Sciences. 23 (2):133-143. doi.org/10.1016/j.ejrs.2019.02.001

We added the reference n.90 in the discussion section, since it demonstrates the
importance of using Google Earth images and tools for detecting land use changes: 

Damtea W, Kim D, Im S. Spatiotemporal analysis of land cover changes in the chemoga
basin, Ethiopia, using Landsat and google earth images. Sustain. 2020;12(9).

Responses to the reviewer’s general observation and questions

1. If the authors have adequately addressed your comments raised in a previous round
of review and you feel that this manuscript is now acceptable for publication, you
may indicate that here to bypass the “Comments to the Author” section, enter your
conflict of interest statement in the “Confidential to Editor” section, and submit
your "Accept" recommendation.

Reviewer #1: All comments have been addressed

Reviewer #2: All comments have been addressed

Reviewer #3: All comments have been addressed

Reviewer #4: All comments have been addressed

Since the reviewers agreed that all comments have been addressed, we have nothing
else to add. 

2. Is the manuscript technically sound, and do the data support the conclusions?

Reviewer #1: Partly

Reviewer #2: Yes

Reviewer #3: Yes

Reviewer #4: Partly

We answered in detailed all the remarks made by the Reviewer#1 in the following
section.

In addition, we have accepted most of the suggestions from the Reviewer#4, which were
included in the document uploaded as attachment; however, they were not substantive
changes. 

Finally, we consider that with the changes made on the previous revised version, we
have improved the theoretical and the methodological framework of the manuscript to
better connect all the elements of our study, fulfilling this evaluation
criterion.

3. Has the statistical analysis been performed appropriately and rigorously?

Reviewer #1: N/A

Reviewer #2: Yes

Reviewer #3: Yes

Reviewer #4: Yes

We have nothing to add to this criterion.

4. Have the authors made all data underlying the findings in their manuscript fully
available?

Reviewer #1: Yes

Reviewer #2: Yes

Reviewer #3: Yes

Reviewer #4: Yes

As it is recognized by all the reviewers, we have made all our data fully available
in the Zenodo’s open access repository. This is the DOI 10.5281/zenodo.5911876 to
access our datasets.

5. Is the manuscript presented in an intelligible fashion and written in standard
English?

Reviewer #1: Yes

Reviewer #2: Yes

Reviewer #3: Yes

Reviewer #4: Yes

We are very pleased that all reviewers agreed that the English grammar of the
manuscript has improved significantly 

Responses to Specific comments from different reviewers:

Specific comments from Reviewer #1:

After reviewing the revised file submitted by the authors, compliance with the
requirements in the initial review phase is generally approved. However, the
following details have been received less attention.

Remote sensing experts believe that accuracy of over 85% is acceptable for detecting
changes in satellite imagery. Hene, reputable papers from remote sensing experts
should be studied in this regard. Assuming that the number and dispersion of
sampling points (Rois) were observed, the accuracy of the article is not
satisfactory. However, with a little leeway, the accuracy of 69-73 and 79 can be
accepted.

As we explained in our previous rebuttal letter, to refine the results and obtain a
more accurate map of the study area, a thorough editing process of the official
vector maps from the different study periods was carried out. To support this
editing process, as carried out by (Madrigal-Martínez and Miralles i García 2019),
distinct secondary sources of information were revised such as field points, Google
Earth images, orthophotographs and other official such as the ecosystem coverage
(http://ide.ambiente.gob.ec/mapainteractivo/), the floriculture
cadastral and other maps from the Ministry of Agriculture (http://geoportal.agricultura.gob.ec/). In
addition, composite LANDSAT images from our study area, using radiometric
enhancements, three-dimensional visualization and spectral band combinations were
also used through Google Earth Engine (Gorelick et al. 2017). From the editing
process, mainly five typologies were improved. These included: planted forests,
developed areas (populated zones), floriculture (areas represented by greenhouses)
and natural water bodies. Following the methods proposed by (Jin et al. 2021), a
point-based accuracy assessment was conducted using Google Earth as a verification
source. 

In the previous version of our manuscript, we added and explanation of our accuracy
assessment in the Methodology as follows:

Following the methods proposed by (Jin et al. 2021), a point-based accuracy
assessment was conducted using Google Earth as a verification source. After that, a
confusion matrix was created using 600 random points obtained from a stratify
sampling scheme over the altitudinal bands. 

The resulting overall accuracy of the edited maps ranged from 82 to 86%; therefore,
we consider we have improved the accuracy to detect changes in our study area close
to the accuracy thresholds suggested by remote sensing experts, as stated by the
Reviewer.

To study the plant and agricultural species, a scale of 1.100000 is small. This scale
is more suitable for maps obtained from low-resolution satellite images and for
studying the regional climate.

The difficulties to detect similar spectral reflectance patterns between neighboring
landscape segments could be overcome by using Google Earth images in a
post-classification process (Damtea, Kim, and Im 2020). In our analysis,
discriminating natural from plantation forests (which are encompassed by different
plant species) could have some of these difficulties. We, therefore, used the
editing or post-classification process described above, using different sources and
the powerful options provided in the Google Earth Engine to improve the
identification of unclear objects (Midekisa et al. 2017; Sidhu, Pebesma, and Câmara
2018). 

Google Earth is not a good source for specific remote sensing tasks. It is more
reliable to use mapping organization maps, aerial mapping, or field fraternity
mapping operations.

We also consider that ground field surveys are preferable to remote sensing data
(such as Google Earth) for accuracy detection or supervised classification uses.
However, recent articles are demonstrating the practicality of using Google Earth
(GE) images when other sources are not available. GE offers the possibility of
conducting LULC analysis with high to medium spatial resolution and with a
multitemporal scope (Damtea, Kim, and Im 2020).

Damtea, Wubeshet, Dongyeob Kim, and Sangjun Im. 2020. “Spatiotemporal Analysis of
Land Cover Changes in the Chemoga Basin, Ethiopia, Using Landsat and Google Earth
Images.” Sustainability (Switzerland) 12(9).

Gorelick, Noel et al. 2017. “Google Earth Engine: Planetary-Scale Geospatial Analysis
for Everyone.” Remote Sensing of Environment 202(2016): 18–27. doi:
10.1016/j.rse.2017.06.031.

Jin, Darren How, Mohd Hasmadi Ismail, Farrah Melissa Muharam, and Mohamad Azani
Alias. 2021. “Evaluating the Impacts of Land Use/Land Cover Changes across
Topography against Land Surface Temperature in Cameron Highlands.” PLoS ONE 16(5
May): 1–26. doi: 10.1371/journal.pone.0252111.

Madrigal-Martínez, Santiago, and José Luis Miralles i García. 2019. “Land-Change
Dynamics and Ecosystem Service Trends across the Central High-Andean Puna.”
Scientific Reports 9(1): 1–12.

Midekisa, Alemayehu et al. 2017. “Mapping Land Cover Change over Continental Africa
Using Landsat and Google Earth Engine Cloud Computing.” PLoS ONE 12(9): 1–15.

Sidhu, Nanki, Edzer Pebesma, and Gilberto Câmara. 2018. “Using Google Earth Engine to
Detect Land Cover Change: Singapore as a Use Case.” European Journal of Remote
Sensing 51(1): 486–500. https://doi.org/10.1080/22797254.2018.1451782.

Specific comments from Reviewer #2:

I have reviewed the manuscript number PONE-D-21-35047 full title ‘Land use and land
cover change in a tropical mountain landscape of northern Ecuador: altitudinal
patterns and driving forces’. The revised manuscript is improved in quality and
readability. On the other hand, I still have few suggestions author/s should take
into consideration:

1. Title should be ‘Land use and land cover change in a tropical mountain landscape
of northern Ecuador’.

We thank the reviewer for this suggestion; however, we prefer the longer title
because we feel that it better describes its content and scope.

2. Abstract: Generalized Additive Model (GAM) (Line 32) and Multiregression models
(GAM) (Line 41) is creating confusion. Please use different abbreviation for two
models.

Completed. As we are referring to the same model in line 32 and line 41 in the
Abstract, we have used the same abbreviation to avoid confusion.

3. Suggested references are missing in the revised manuscript reference section
kindly add for value adition.

• Mishra, P.K. Rai, A. and Rai, S.C. (2020) Land Use and Land Cover Change Detection
using Geospatial Techniques in Sikkim Himalaya, India. Egyptian Journal of Remote
Sensing and Space Sciences. 23 (2):133-143. doi.org/10.1016/j.ejrs.2019.02.001

• Mishra, P.K. and Rai, A. (2020) Role of Unmanned Aerial Systems for Natural
Resource Management. Journal of the Indian Society of Remote Sensing. ISSN
0974-3006. DOI: 10.1007/s12524-020-01230-4

We have complemented our references with the first suggested article. However, we
feel that the second one does not relate to the main methodology used in our
manuscript, therefore, we did not included it.

4. The practical implacability of the study in any other field of work is still
missing. Recommendations for Future Work may add value to the manuscript.

Best Wishes!

We don’t fully understand this suggestion because our revised manuscript already
includes a section for implications (from line 691 to 721); however, we added
another sentence to highlight the implication of our findings for conserving native
ecosystems (lines 716-717). 

We deleted the subtitle of limitations because the last paragraphs before the
conclusions described the implications of our study.

Finally, at the end of the conclusion section, we added our ideas for future
work.

Reviewer #3: The authors have provide enough explanation to my concern. I still
wished they had the latest year covered in the analysis but I do not seek that as
the sole reason to reject it. Hence, I recommend for acceptance to publish provided
that it meets other publication criteria as per the guideline ( copy right and
others).

We thank the reviewer#3 for recommending the acceptance of our manuscript in PLOS
ONE. 

As we explained in our previous rebuttal letter. We decided to use official LULC maps
and other freely available databases from distinct sources (demographic, climatic,
topographic, etc.) as a practical DPSIR implementation example that could be
replicated to understand environmental change (LULC transitions and their drivers)
in tropical mountain systems. These regions, mostly situated in developing
countries, are characterized by a context of data scarcity and low technical
capacities for the processing of remote sensing information required for land
management and planning.

Specifically, this situation depicts the situation from our study region.

For our LULC analysis we only included the four periods of time 1990, 2000, 2008 and
2014 from which available and comparable maps were available when the analysis took
place.

Reviewer #4: Dear Authors,

Thank you for your manuscript “Land use and land cover change in a tropical mountain
landscape of northern Ecuador: altitudinal patterns and driving forces”. The title
is now well written covering the content of the paper. I recommended accepting the
topic as it fully matches the scope of the journal “PLOS ONE”. Also, now authors
have incorporated most reviewers’ comments provided to them making the manuscript
much better. But there are some issues that need to be addressed by the authors, see
the attached track change word document.

We thank the reviewer#4 for recommending the acceptance of our manuscript in PLOS
ONE.

We have accepted most of the suggestions made by the reviewer in the manuscript, they
were mainly changes in rephrasing or shortening the sentences. Only one suggestion
was not accepted, it was the deletion of S2Fig; we consider important to keep it as
it is, since it complements the analysis of native ecosystems in different elevation
bands. We rewrote the legend title of S1Fig and S2Fig.

We look forward to hearing from you in due time regarding our submission and to
respond to any further questions and comments you may have. 

Sincerely,

Paulina Guarderas, M. Sc.

Corresponding Author

Facultad de Ciencias Biológicas,

Universidad Central del Ecuador

Numa Pompilio Liona S/N y Yaguachi,

QUITO, Ecuador

E-mail: apguarderas@uce.edu.ec

Cel: 593 992403093

 Additional Contact:

Biodiversity and Landscape Unit,

Gembloux Agro Bio-Tech,

University of Liège

Passage des Dport s 2, 5030

GEMBLOUX, Belgique

E-mail: ap.guarderas@doct.uliege.be

to Reviewers.docx
---

## [Editor Report · Decision Letter 2]

17 May 2022

PONE-D-21-35047R2Land use and land cover
change in a tropical mountain landscape of northern Ecuador: altitudinal patterns
and driving forcesPLOS ONE

Dear Dr. Guarderas,

Thank you for submitting your manuscript to PLOS ONE. After careful consideration, we
feel that it has merit but does not fully meet PLOS ONE’s publication criteria as it
currently stands. Therefore, we invite you to submit a revised version of the
manuscript that addresses the points raised during the review process.

Please submit your revised manuscript by Jul 01 2022 11:59PM. If you will need more
time than this to complete your revisions, please reply to this message or contact
the journal office at plosone@plos.org. When
you're ready to submit your revision, log on to https://www.editorialmanager.com/pone/ and select the 'Submissions
Needing Revision' folder to locate your manuscript file.

Please include the following items when submitting your revised
manuscript:A rebuttal letter that responds to each point raised by the academic
editor and reviewer(s). You should upload this letter as a separate file
labeled 'Response to Reviewers'.A marked-up copy of your manuscript that highlights changes made to the
original version. You should upload this as a separate file labeled
'Revised Manuscript with Track Changes'.An unmarked version of your revised paper without tracked changes. You
should upload this as a separate file labeled 'Manuscript'.If you would like to make changes to your financial disclosure,
please include your updated statement in your cover letter. Guidelines for
resubmitting your figure files are available below the reviewer comments at the end
of this letter.

We look forward to receiving your revised manuscript.

Kind regards,

Manoj Kumar

Academic Editor

PLOS ONE

Journal Requirements:

Additional Editor Comments:

All of the comments have been addressed while there remains some queries to be looked
before final acceptance. Kindly address the queries raised by the reviewer (although
minor one) for further necessary action.
---

## [Author Response · Author response to Decision Letter 2]

14 Jun 2022

Since the reviewers raised no further observations or comments on the latest version
of the manuscript, my co-authors and I have concentrated on thoroughly revising the
citations and bibliographic references so that they comply with the citation style
of the journal and contain complete information, as it was requested in your last
message.

Likewise, we have reviewed in detail which references have been retracted or replaced
in the manuscript text and the Reference list to provide the rationale for doing
so.

to Reviewers.docx
---

## [Editor Report · Decision Letter 3]

28 Jun 2022

Land use and land cover change in a tropical mountain landscape of northern Ecuador:
altitudinal patterns and driving forces

PONE-D-21-35047R3

Dear Dr. Guarderas,

We’re pleased to inform you that your manuscript has been judged scientifically
suitable for publication and will be formally accepted for publication once it meets
all outstanding technical requirements.

Kind regards,

Manoj Kumar

Academic Editor

PLOS ONE
---

## [Editor Report · Acceptance letter]

4 Jul 2022

PONE-D-21-35047R3 

Land use and land cover change in a tropical mountain landscape of northern Ecuador:
altitudinal patterns and driving forces 

Dear Dr. Guarderas:

I'm pleased to inform you that your manuscript has been deemed suitable for
publication in PLOS ONE. Congratulations! Your manuscript is now with our production
department. 

Kind regards, 

on behalf of

Dr. Manoj Kumar 

Academic Editor

PLOS ONE